# Attenuation of dopamine-modulated prefrontal value signals underlies probabilistic reward learning deficits in old age

Lieke de Boer[1]*, Jan Axelsson[2,3], Katrine Riklund[2,3], Lars Nyberg[2,3,4], Peter Dayan[5], Lars Bäckman[1], Marc Guitart-Masip[1,6]*

[1]Aging Research Center, Karolinska Institute, Stockholm, Sweden; [2]Department of Radiation Sciences, Diagnostic Radiology, Umeå University, Umeå, Sweden; [3]Umeå Center for Functional Brain Imaging, Umeå University, Umeå, Sweden; [4]Department of Integrative Medical Biology, Physiology, Umeå University, Umeå, Sweden; [5]Gatsby Computational Neuroscience Unit, University College London, London, United Kingdom; [6]Max Planck UCL Centre for Computational Psychiatry and Ageing Research, University College London, London, United Kingdom

*For correspondence:
liekelotte@gmail.com (LB);
marc.guitart-masip@ki.se (MG-M)

Competing interests: The authors declare that no competing interests exist.

**Abstract** Probabilistic reward learning is characterised by individual differences that become acute in aging. This may be due to age-related dopamine (DA) decline affecting neural processing in striatum, prefrontal cortex, or both. We examined this by administering a probabilistic reward learning task to younger and older adults, and combining computational modelling of behaviour, fMRI and PET measurements of DA D1 availability. We found that anticipatory value signals in ventromedial prefrontal cortex (vmPFC) were attenuated in older adults. The strength of this signal predicted performance beyond age and was modulated by D1 availability in nucleus accumbens. These results uncover that a value-anticipation mechanism in vmPFC declines in aging, and that this mechanism is associated with DA D1 receptor availability.
DOI: https://doi.org/10.7554/eLife.26424.001

## Introduction

In order to navigate an uncertain world successfully, humans and other animals are required to learn and update the values of available actions and switch between them appropriately. Compared with younger adults, older individuals are poor at probabilistic reward learning and subsequent optimal action selection (*Eppinger et al., 2011*; *Mell et al., 2005*). One common account of this deficit is an age-related deterioration of the dopamine (DA) system (*Volkow et al., 1998*), with two of its primary targets - striatum and prefrontal cortex (PFC) - being obvious culprits.

A wealth of animal literature demonstrates that DA signals from midbrain convey reward prediction errors (RPEs) (*Bayer and Glimcher, 2005*; *Schultz et al., 1997*), which are thought to act as signals that facilitate action selection in striatum (*Niv and Montague, 2009*; *Pessiglione et al., 2006*). Hence, one hypothesis states that aging leads to decreased striatal DA release in response to RPEs, leading to a comparatively less efficient learning signal (e.g. slower learning rate). Supporting this, previous studies reported lower correlations between RPEs generated from probabilistic reward learning tasks and nucleus accumbens (NAcc) BOLD signals in older compared with younger adults (*Eppinger et al., 2013*; *Samanez-Larkin et al., 2014*). By decomposing RPEs in a dynamic two-armed bandit task into their two subcomponents: obtained reward (R) and expected value (Q), *Chowdhury et al. (2013)* showed that, in older adults, neural activity in NAcc reflected just the

former. Only after the dopaminergic system had been pharmacologically boosted could the expected value component be detected in NAcc. While these findings support the tenet that attenuation of DA-modulated expected value signals in NAcc underlies age-related performance deficits (*Chowdhury et al., 2013*), a younger comparison group was lacking.

Another hypothesis is that age-related decline in probabilistic reward learning may be related to impaired prefrontal functioning (*Nyberg et al., 2010*; *Raz et al., 2005*; *Halfmann et al., 2016*). Indeed, compromised DA projections to frontostriatal circuits are reported in aging (*Dreher et al., 2008*; *Hämmerer and Eppinger, 2012*). Anticipatory activity reflecting the value of the chosen option in the ventromedial PFC (vmPFC) is widely reported in decision-making tasks (*Balleine and O'Doherty, 2010*; *Daw et al., 2006*) and is modulated by DA (*Jocham et al., 2011*). Supporting an involvement of PFC in age-related decline in probabilistic reward learning, one previous study suggests decreased RPE signalling in vmPFC in older adults (*Eppinger et al., 2015*). Another study showed that within a group of older adults with increased BOLD activity related to value anticipation predicted better performance on the Iowa gambling task (*Halfmann et al., 2016*). However, despite evidence suggesting that age-related decline in PFC value signals could be related to dopaminergic deterioration, there is no published data directly showing this.

Furthermore, there is little work comparing younger and older populations according to an additional factor that could influence performance in these tasks, namely the impact of uncertainty or confidence in the payoffs or values of the options on choice switching (*Badre et al., 2012*; *Frank et al., 2009*; *Vinckier et al., 2016*). Uncertainty should influence the trade-off between exploration and exploitation (*Sutton and Barto, 1998*) that an optimal policy should balance. However, how exploration and switching are modulated in aging and how they influence performance is unclear.

Our aim was to investigate the effect of age and DA availability on striatal and prefrontal mechanisms involved in probabilistic reward learning. We included samples of 30 older and 30 younger participants who performed a two-armed bandit task (TAB) previously used by *Chowdhury et al. (2013)* while fMRI data was acquired. All participants were healthy and cognitively high functioning (MMSE > 27). In brief, all participants performed 220 trials on the TAB (*Figure 1a*). On each trial, participants chose between one of two bandits, represented by fractal images. After a variable interval, the outcome was presented as a green arrow pointing up signalling a reward, or a yellow horizontal bar signalling no reward. Probabilities of obtaining a reward varied over time for both bandits, according to independent Gaussian random walks (*Figure 1c*, left). This required the participants to update the expected value for each bandit continuously. Participants received monetary earnings of 1 Swedish Krona (SEK, ~$0.11) per rewarded trial. Behaviour was quantified with a Bayesian observer model augmented to capture the influence of variance and confidence on switch behaviour. This model outperformed a Rescorla-Wagner (RW) model that tracked expected value using simple RPEs. To investigate the relationship between the ability to learn about probabilistic rewards and the DA system, we collected PET data using the radioligand [11C]SCH23390 to measure striatal and prefrontal DA D1 receptor binding potential (D1 BP), as a proxy for integrity of the dopaminergic system. The chosen radioligand allows for reliable measurement of BP in striatum and PFC simultaneously (*Hall et al., 1994*), as opposed to alternative markers of dopaminergic function.

Based on previous work, we hypothesised that, in younger participants, BOLD signal in NAcc would reflect both components of the RPE signal, whereas older participants would show a reduced expected-value component. Additionally, we expected an attenuated expected-value signal during choice in the older compared to the younger sample in PFC. We reasoned that the strength of these expected-value representations in both PFC and NAcc would show a relationship to DA D1 BP in either subcortical or prefrontal regions.

## Results

### Task performance

The goal of the analyses was to establish the neural mechanism underlying decreased probabilistic value learning in older participants. We did this by (1) assessing differences between age groups in the BOLD signal related to anticipatory expected value in the vmPFC, (2) assessing differences between age groups in the BOLD signal related to RPEs in the NAcc, and (3) investigating the

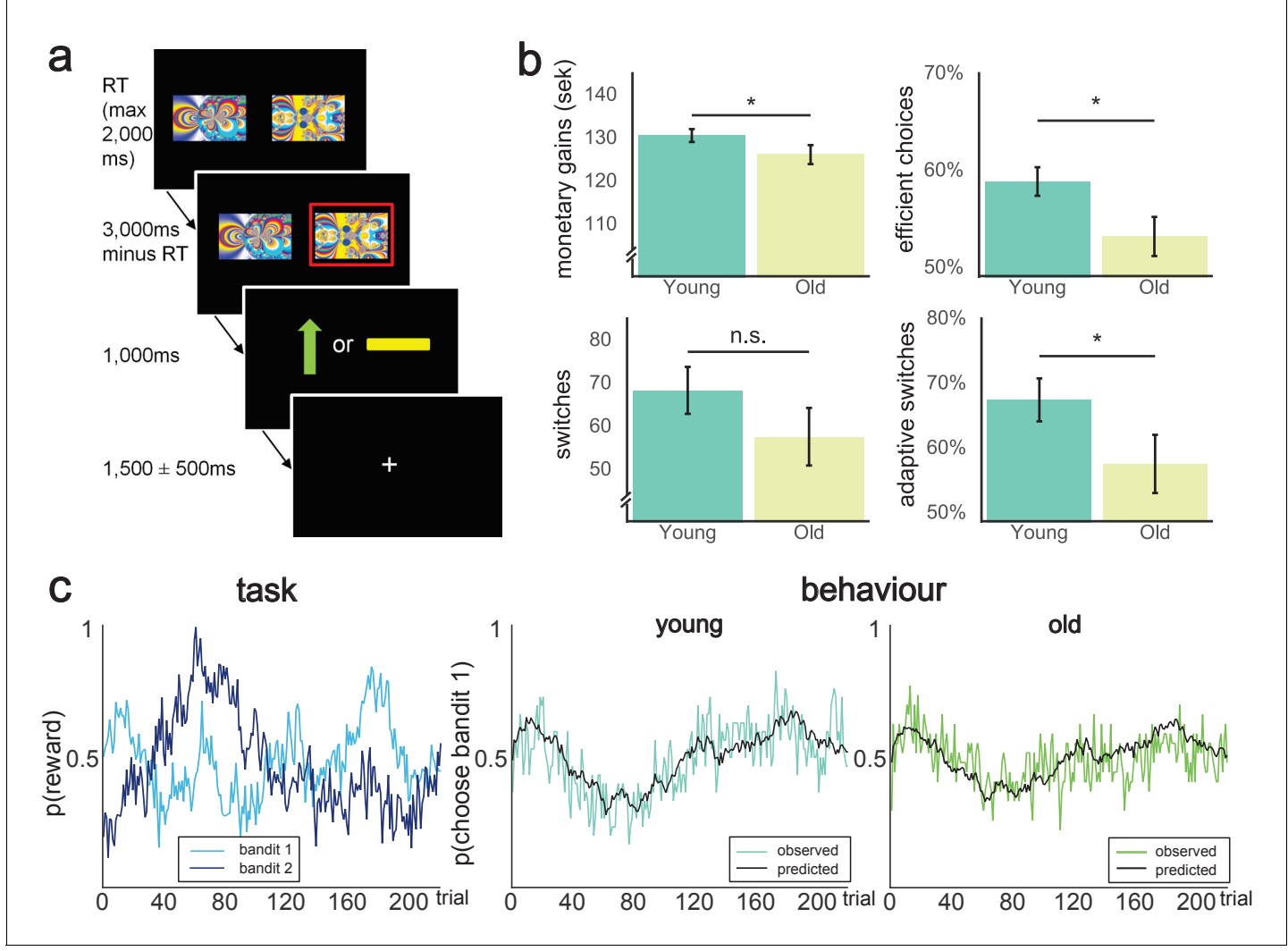

**Figure 1.** Behavioural paradigm and performance on the two-armed bandit task. (a) Schematic representation of a trial in the TAB. Participants were presented with two fractal images on each trial and selected one of them through a button press. The maximum response time was 2000 ms, meaning the trial would count as a miss if the response time exceeded this limit and the next trial would start immediately after the next inter-trial interval. If one stimulus was selected, this option was highlighted with a red frame. After 1000 ms, participants were presented with the outcome: either a green arrow pointing upwards, indicating an obtained reward of 1SEK (≈ $0.11), or a yellow horizontal bar, indicating no win. The position of the images on the screen varied randomly across the 2 × 110 trials of the experiment. Reward probabilities varied throughout the experiment. (b) Behavioural performance on the TAB, across age group. Younger participants earned more money on the TAB on average (top left, t(49) = 1.69, p(one-tailed) =0.048). Proportion of efficient choices differed significantly between the two groups (top right, Mann-Whitney U = 286.5, p(one-tailed)=0.029). Number of switches did not differ significantly between groups (p=0.19; bottom left), but the proportion of adaptive switches differed between age groups (bottom right, Mann-Whitney = 271.0; p(two-tailed)=0.033). Data are represented as mean ±SEM. (c) Left pane: Varying reward probabilities for obtaining a reward for each bandit on the 220 trials of the experiment. Center/right pane: Model predictions (black lines) and observed behaviour (coloured lines). Model fit did not significantly differ between participants (Mann-Whitney U = 353.0, p=0.406).

DOI: https://doi.org/10.7554/eLife.26424.002

The following source data, source code and figure supplements are available for figure 1:

**Source data 1.** Source data to *Figure 1*.

DOI: https://doi.org/10.7554/eLife.26424.005

**Source code 1.** Code that was used to perform simulation of behavioural data (figure 1c), as well as the creation of *Figure 1*.

DOI: https://doi.org/10.7554/eLife.26424.006

**Figure supplement 1.** Dopamine D1 binding potential is lower in older adults.

DOI: https://doi.org/10.7554/eLife.26424.003

**Figure supplement 1—source data 1.** Binding potentials in seven ROIs for young and old participants.

DOI: https://doi.org/10.7554/eLife.26424.004

relationship between these BOLD signals and DA D1 binding potentials (BP) in a set of predefined ROIs. To obtain the best estimate of expected value to use in our fMRI analysis, we fitted a range of computational models and used Bayesian model selection.

Younger adults outperformed older adults on the task (*Figure 1b*). There was a weak group difference in total amount of money earned ($M_{old}$ = 125.9, SD = 11.4; $M_{young}$ = 130.3, SD = 8.2; t(49), p(one-tailed)=0.050). Additionally, efficient choices, defined as the proportion of total choices that were more likely to be rewarded according to the actual (hidden) state of the Gaussian random walks also differed between groups ($M_{old}$ = 0.53, SD = 0.10; $M_{young}$ = 0.59. SD = 0.08, Mann-Whitney U = 286.5, p(one-tailed)=0.029. We also investigated how switching between the two alternatives contributed to performance. The number of switches was negatively related to total monetary gains (r = −0.29, p=0.032, controlled for age). There was no evidence to suggest that the number of switches differed between age groups ($M_{old}$ = 57.3, SD = 34.6; $M_{young}$ = 68.1. SD = 29.8, Mann-Whitney U = 323.0, p=0.190).

We assessed the ability of a variety of members of two broad families of models to capture trial-by-trial behaviour (see Materials and methods, SI for details). The first family includes variations on standard reinforcement learning (RL) models in which action values are learned through RPEs and the RW updating rule. The second family of models comprises variations on a Bayesian observer in which the probability distribution of obtaining a reward is updated after each outcome observation. Model comparison statistics are displayed in *Table 1*.

The most parsimonious account came from a five parameter Bayesian observer model. This tracked the probability of obtaining a reward for each action as a beta distribution with parameters representing pseudo-counts of wins and win omissions. Pseudocounts for the bandit that was chosen were updated according to the outcome, based on a learning rate of ω. Pseudocounts for the bandit that was not chosen were relaxed towards neutral values based on a forgetting rate of λ. The beta distributions generated action propensities for the two bandits according to three weighted additive factors: one was the relative expected values (Q) of the bandits, calculated as the mean of the beta distributions (*Figure 2* and *Equation 7*, Materials and methods). The other two depended on different forms of uncertainty and were associated with the choice between sticking with the previous choice or switch.

The first determinant of switching was the current variance (V) of the option that was not chosen on the previous trial calculated from its approximate beta distribution (*Figure 2*; formula 8, Materials and methods). The variance was multiplied by a parameter υ and added to the propensity of this previously unchosen option. υ was negative in all but two participants (*Table 2*), reflecting the fact that increasing uncertainty about the unchosen option decreased its value, making it a less likely

**Table 1.** Model comparison statistics for the different models.

The winning model, defined as the model with the lowest integrated BIC (iBIC), was the Bayesian observer model with five parameters. Parameters: β: inverse temperature parameter for softmax, α: learning rate for RW model, b: choice kernel, ϕ: forgetting rate for RW model, ω: learning rate for Bayesian model, λ: forgetting rate for Bayesian model, υ: variance weighting, κ: confidence weighting.

| Family | Parameters | # Param | Likelihood | Pseudo-$R^2$ | iBIC |
|---|---|---|---|---|---|
| RW | β, α | 2 | −5636.8 | 0.336 | 11309 |
| | β, α, b | 3 | −5317.8 | 0.374 | 10692 |
| | β, α, b, ϕ | 4 | −5140.0 | 0.394 | 10355 |
| Bayesian observer | β, ω | 2 | −5919.8 | 0.302 | 11877 |
| | β, ω, λ | 3 | −5719.2 | 0.326 | 11495 |
| | β, ω, λ, b | 4 | −5154.6 | 0.392 | 10385 |
| | β, ω, λ, υ(chosen) | 4 | −5161.7 | 0.392 | 10399 |
| | β, ω, λ, υ(unchosen) | 4 | −5130.0 | 0.395 | 10335 |
| | β, ω, λ, κ | 4 | −5675.3 | 0.331 | 11426 |
| | β, ω, λ, υ(unchosen), κ | 5 | −5082.5 | 0.401 | 10259 |

DOI: https://doi.org/10.7554/eLife.26424.007

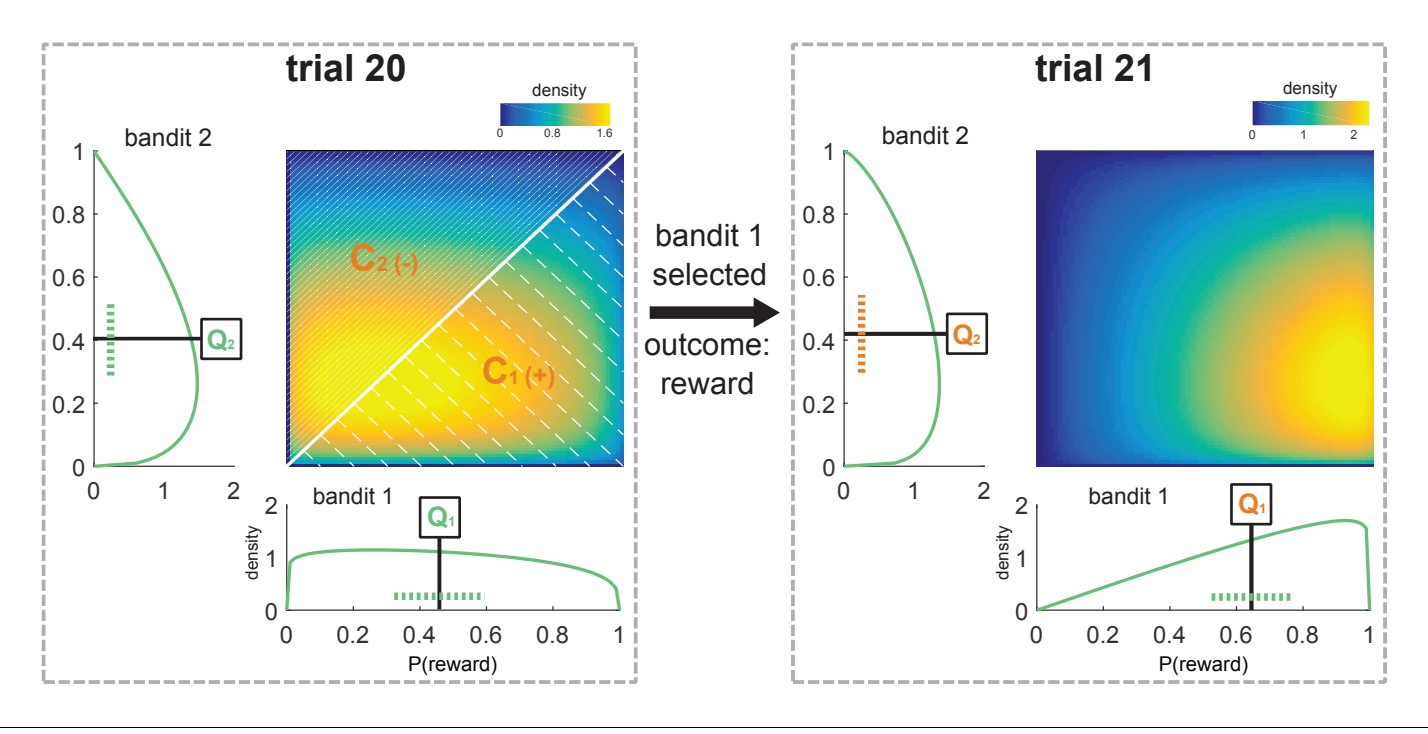

**Figure 2.** Schematic representation of the Bayesian model values for one participant at the time of choice at trial 21. All components that are used to model choice at trial 21 are marked in orange. The sequence of choices for this participant was [1 1 1 2 1 1 1 2 2 1 2 2 1 1 1 2 2 2 2 1], and the payout for these choices was [1 1 0 0 1 1 0 1 0 0 0 0 1 1 0 0 1 1 0 1]. According to the participant's individually fitted model parameters ($\omega$ = 0.72; $\lambda$ = 0.28), and following this sequence of choices and outcomes, the beta distributions defining the subjective value of the bandits were $\theta_1 \sim \beta(\theta_1; 2.02, 1.08)$ and $\theta_2 \sim \beta(\theta_2; 1.26, 1.74)$ (see *Equations 9–11*, Materials and methods) at choice of trial 21. The expected value for each bandit was defined as the mean of the beta distribution ($Q_1$ = 0.65, $Q_2$ = 0.42; see *Equation 7*, Materials and methods). The variance of the unchosen option was equal to the variance of bandit 2, which was not chosen on trial 20 ($V_{uc}$ = 0.05, see *Equation 8*, Materials and methods). Variance is schematically represented as a dotted line (note that this is an approximation because the beta distributions are not symmetrical). The 2-d plot shows the joint distribution $P(\theta_1,\theta_2)$ where values of $\theta_1$ are along the x-axis and $\theta_2$ along the y-axis. Confidence was calculated based on the values of the distributions at choice on the previous trial. $C_1$ was defined as the probability that a random sample drawn from $\theta_1$ at the time of choice at trial 20 was greater than a sample drawn from $\theta_2$ (shaded area below the diagonal, as $\theta_1 > \theta_2$ there. $C_1$ = 0.56, Materials and methods *Equation 15*). $C_2$ could be defined as $1-C_1$ (shaded area above the diagonal, $C_2$ = 0.44, *Equation 16*, Materials and methods). $C^{rel}$ was equivalent to $C^{chosen} - C^{unchosen}$, in this case $C_1-C_2$ ($C^{rel}$ = 0.12, *Equation 17*). This relative confidence was scaled by $\kappa$ and then added to the action that was not chosen on the previous trial (in this case bandit 2).

DOI: https://doi.org/10.7554/eLife.26424.008

choice on the current trial. Hence, increased uncertainty about the previously unchosen option caused most subjects to stick to their current choice. This is the opposite of an exploration bonus. This model outperformed an account based on a more conventional choice kernel, according to which perseverating or switching was influenced only by previous choices themselves rather than something reflecting knowledge about those choices. This suggests that perseveration, which is

**Table 2.** Summary statistics of the five parameters of the winning model.

|  | Minimum | 25th percentile | Median | 75th percentile | Maximum |
|---|---|---|---|---|---|
| $\beta$ | 1.436 | 7.017 | 12.280 | 17.730 | 64.750 |
| $\omega$ | 0.042 | 0.238 | 0.408 | 0.558 | 0.851 |
| $\lambda$ | 0.055 | 0.139 | 0.202 | 0.270 | 0.544 |
| $\upsilon$ | −3.372 | −1.845 | −1.069 | −0.678 | 0.234 |
| $\kappa$ | −0.202 | 0.152 | 0.260 | 0.359 | 0.896 |

DOI: https://doi.org/10.7554/eLife.26424.009

commonly observed (*Rutledge et al., 2009*; *Schönberg et al., 2007*), may partly reflect uncertainty aversion.

The second determinant of switching was a measure of the relative confidence in the choice that was made on the previous trial (see Materials and methods). We assumed that subjects used their approximate Bayesian posterior distributions over the values of the bandits to calculate this confidence, $C^{rel}$, as their subjective probability that the option they chose was better (a calculation they made before observing the outcome on that trial). A term $C^{rel}\kappa$ was then added to value of the action that was not chosen on that previous trial (see Methods) where $\kappa$ was fitted to each participant. Thus, if $\kappa$ was positive, then a subject would be more likely to switch on trial $t$ if she had been more confident on trial $t - 1$.

Note that relative confidence was calculated on the preceding trial because, at the time of the choice, the model has no information about the option that will be chosen on that trial. Perhaps surprisingly, $\kappa$ was positive in 49 out of 57 participants (*Table 2*) – thus for the majority of subjects, the more sure they were that the chosen option was better, the more they sought to switch and try the alternative. It is important to acknowledge, however, that there are subtle interactions with the effects of the means and variances of the options with which relative confidence is partly correlated. Nevertheless, $\kappa$ was negatively correlated with total monetary gains on the task (r(54) = 0.42, p=0.001, controlled for age; *Supplementary file 1*), with negative values of $\kappa$ in those participants with the highest performance. This implies that $\kappa$ has the expected effect on performance despite having an unexpected sign at the group level. The overall tendency for $\kappa$ to be positive and $\upsilon$ to be negative does not stem from autocorrelation between the two as the sign of these parameter is largely the same when the model is specified with only one of these parameters (data not shown).

The final step to realizing choice was to feed the ultimate action propensities into a softmax with temperature parameter $\beta$.

We found that variation in the number of switches was better accounted for by variation in the parameters of the winning model than by that in parameters of the best RW model (*Supplementary file 1b*). No single model parameter differed between age groups (Mann-Whitney test: $\beta$, U = 386.0, p=0.761; $\omega$, U = 345.0, p=0.338; $\lambda$, U = 401.0, p=0.949; $\upsilon$, U = 307.0, p=0.117; $\kappa$, U = 374.0, p=0.620). A multivariate analysis with the model parameters as independent variables and age group as a fixed factor did not yield any significant predictor of age group (F = 0.91, p=0.482). Model fit, defined by the individual log likelihood for each participant, also did not differ between age groups (Mann-Whitney U = 353.0, p=0.406). In the best-performing RW model (which fit the behavioural data less well), younger participants learned more quickly (*Supplementary file 1c*).

Because measures of successful performance differed between groups but the number of switches did not, we used our winning model to investigate the nature of switches separately in each group. We used the expected values from the winning model to assess the proportion of adaptive switches (to subjectively better bandits) versus maladaptive switches (to subjectively worse bandits) and found that young participants had a higher proportion of adaptive switches compared with old participants ($M_{old}$ = 57.4, SD = 23.4; $M_{young}$ = 68.3. SD = 18.2, Mann-Whitney test U = 271.5, p=0.033, *Figure 1b*, bottom right). The proportion of adaptive switches was positively associated with total monetary gains (r(54) = 0.49, p<0.001), suggesting that the age difference in performance partly resulted from differences in strategic switching.

## Value anticipation in vmPFC

To investigate brain activity reflecting value anticipation, we estimated a GLM that included the chosen value Q on that trial as a regressor to be correlated with the BOLD signal at the time of choice (see Materials and methods, GLM 1). Clusters in vmPFC, bilateral hippocampus, visual cortex and bilateral precuneus showed a positive correlation between BOLD and Q at the time of choice at p (FWE-corrected)<0.05 (*Supplementary file 3*).

We next tested if the expected-value signal differed between age groups. We used the cluster showing a positive correlation between BOLD and Q at the time of choice in vmPFC (*Figure 3*, *Supplementary file 3*) as a functional ROI to extract the individual parameter estimates. A two-sample t-test, orthogonal to the test used to define this ROI, revealed that younger participants showed a stronger representation of Q in vmPFC compared to the older participants ($M_{old}$ = 2.84,

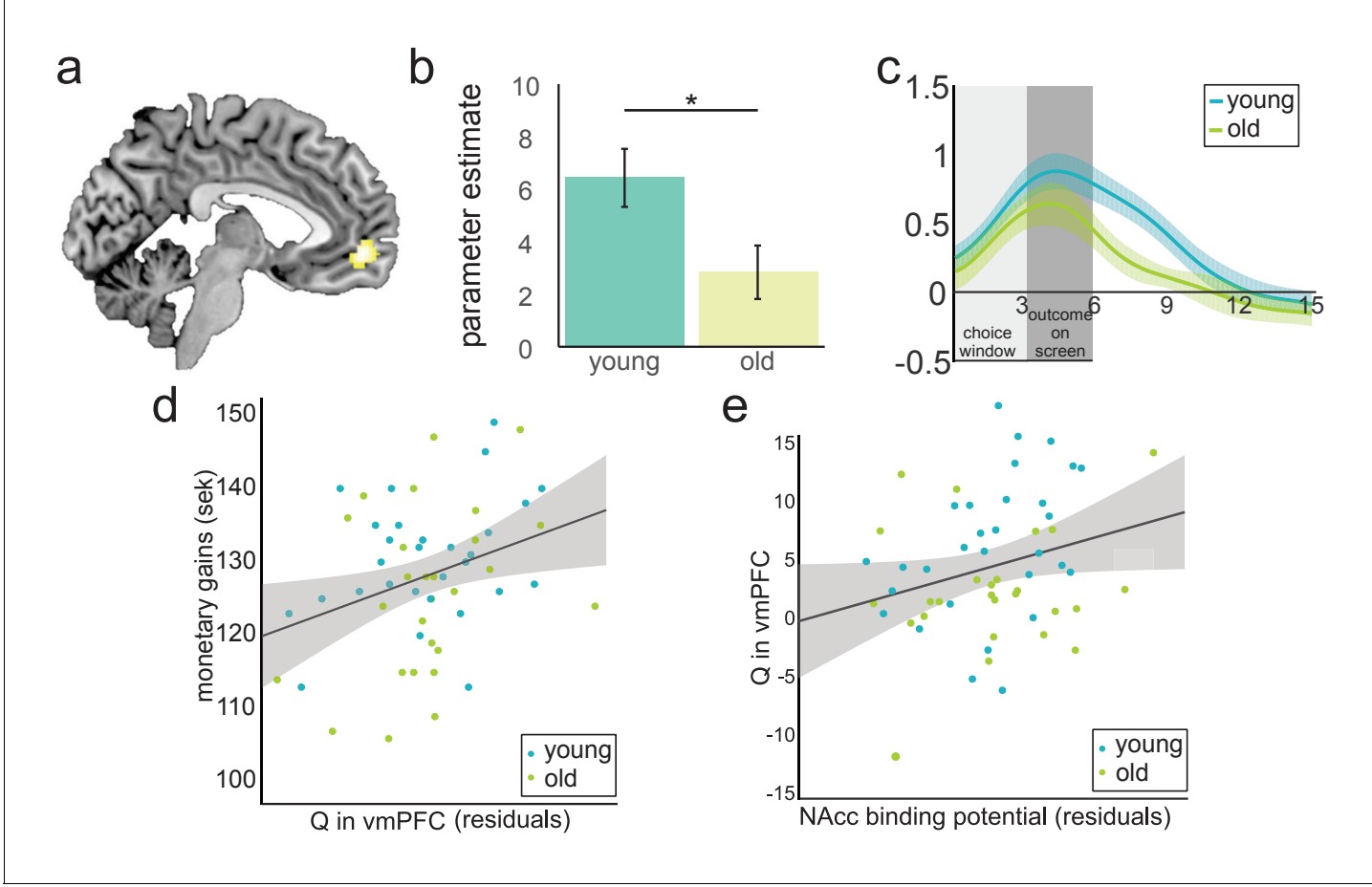

**Figure 3.** Value anticipation in vmPFC is related to behavioural performance and D1 BP in NAcc. (a) Cluster in vmPFC that shows expected value activity at the time of the choice. Peak voxel x,y,z −5,52,–6; p<0.05, FWE corrected. (b) Parameter estimates for younger and older participants extracted from the cluster in *Figure 3a*. Activity differs significantly between age groups (t(55) = 2.38; p=0.021). Error bars represent standard errors of the means. (c) Time-course visualisation of the expected value signal in vmPFC. Shaded areas indicate standard errors. The expected-value signal is significantly larger and prolonged in the younger compared to the older sample. (d) There is a positive relationship between expected-value signal magnitude and total monetary gains (r(53) = 0.37, p=0.006 when controlling for age and model fit). For display purposes, the correlations are shown with residuals after regressing out age and model fit. (e) DA D1 BP in NAcc is positively related to Q in vmPFC (r(53) = 0.28, p=0.038, when controlling for age). For display purposes the correlations are shown with residuals after regressing out age.

DOI: https://doi.org/10.7554/eLife.26424.010

The following source data is available for figure 3:

**Source data 1.** Source data for figure 3: cluster corresponding to Q in vmPFC at the time of choice.

DOI: https://doi.org/10.7554/eLife.26424.011

SD = 5.25; $M_{young}$ = 6.44, SD = 6.07; t(55) = 2.38; p=0.021). This difference in vmPFC value signal did not arise because of the difference in learning performance: when we restricted our analysis to high performers as defined by a median split (13 old, 15 young), a difference in performance was no longer significant (p=0.60), but the strength of expected-value signal in vmPFC was correlated with age (r(26) = −0.39, p=0.040) and we found a marginally significant difference between age groups ($M_{old}$ = 4.21, SD = 4.81; $M_{young}$ = 8.29, SD = 5.72; t(26) = 2.03, p=0.054). For illustrative purposes, we plotted the time course of the expected-value signal in vmPFC over the course of a trial. This suggests that, on average, the expected-value signal was stronger and sustained for longer throughout the trial in younger compared with older adults (*Figure 3c*).

The parameter estimate for Q in vmPFC was positively related to total monetary gains (r(53) =0.37, p=0.006, controlling for age and model fit in a partial correlation). This correlation remained significant without controlling for age, model fit or both. Q in vmPFC was a significant predictor of

all measures of performance (bivariate correlations: total monetary gains: r(55) = 0.47, p<0.001 adaptive switches: r(55) = 0.39, p=0.003; efficient choices: r(55) = 0.38, p=0.004). Age was also a significant predictor of all measures of performance (bivariate correlations: total monetary gains: r(55) = -0.32, p=0.050; adaptive switches: r(55) = -0.26 p=0.052; efficient choices: r(55) = -0.32 p=0.015). Age was also a significant predictor of Q in vmPFC (r(55) = -0.32 p=0.016). Age was no longer a significant predictor of performance after controlling for Q in vmPFC (beta age = −0.12,–0.23, and −0.15; p=0.328, 0.086 and 0.255, for monetary gains, efficient choices and adaptive switches, respectively), whereas Q in vmPFC remained a significant predictor of all measures of performance (beta Q in vmPFC = 0.43, 0.30, and 0.34; p=0.001, 0.023 and 0.012 for monetary gains, efficient choices and adaptive switches, respectively). This is consistent with a full mediation of age effects on performance by Q in vmPFC. Note, however, that it is difficult to make inferences on mediation effects of age in a cross-sectional dataset (*Lindenberger et al., 2011*).

The results were not dependent on the use of the Bayesian model to estimate Q values (when using the RW model Q estimates; when including both age and Q, beta age = −0.20,–0.22, −0.21, p=0.111, 0.093, 0.104; beta Q in vmPFC = 0.33, 0.28, 0.26, p=0.010, p=0.030, p=0.047 for monetary gains, efficient choices and adaptive switches, respectively).

## RPE signals in striatum

RPEs are widely reported in NAcc (*Behrens et al., 2008*; *Niv et al., 2012*); but see also (*Stenner et al., 2015*; *Wimmer et al., 2014*). RPEs are thought to be a critical signal conveyed by dopaminergic neurons (*Bayer and Glimcher, 2005*; *Hart et al., 2014*) that guide action selection in probabilistic learning tasks (*Pessiglione et al., 2006*; *Hart et al., 2014*; *Rolls et al., 2008*) like the TAB. Although our winning computational model, a Bayesian observer model, does not use RPEs, we may expect the brain to, nonetheless, track RPEs as the discrepancy between outcomes observed and outcomes predicted by the model (*Daw et al., 2011*). When investigating RPE signals in fMRI data, a common approach is to identify regions in which activity is correlated with the RPE defined as a single regressor $(R(t)–Q_a(t))$. However, because R and RPE are correlated (*Behrens et al., 2008*; *Niv et al., 2012*; *Li et al., 2011*), when using this approach the amount of variance attributed to RPE may be overestimated (*Behrens et al., 2008*; *Guitart-Masip et al., 2012*) and the identified signals can be seen as putative RPEs. For this reason, it has been suggested that the effects of R and Q need to be estimated separately and only regions showing both signals with opposite signs can be considered as conveying a canonical RPE signal (*Behrens et al., 2008*).

Following this approach, we first defined an ROI for NAcc in each hemisphere in which BOLD was correlated with the full RPE regressor at the time of outcome (Materials and methods, GLM 2, *Figure 4a*, MNI peak voxel coordinates x,y,z = 14,12,–10; k = 72; z = 7.03 and x,y,z = -14,8,–10; k = 47; z = 6.74 with p(FWE-corrected)<0.05). From these regions, we extracted parameter estimates for reward and expected value separately as estimated in a separate GLM model (Materials and methods, GLM 3). We replicated previous findings in older adults (*Chowdhury et al., 2013*), as we saw a significant effect of R, but no significant negative effect of Q in both ROIs (*Figure 4b*). Contrary to our hypothesis, we did not observe a canonical prediction error in the young sample either. Again, we observed a positive effect of R, but no significantly negative effect of Q. Note that this is not inconsistent with the result reported by *Chowdhury et al. (2013)*, where no fMRI data were collected for the young control group. No evidence for differences between the different age groups' mean activation for R or Q were found (p>0.29). In addition, when performing a less stringent test and extracting parameter estimates from this ROI for the full RPE, defined as one regressor (R-Q), we did not observe any differences between the groups' mean activation (p>0.45). These negative results were not dependent on using the Bayesian observer model to generate Q as they were consistent across models (Supporting figure to *Figure 4*). There was no indication that the lack of expected value signal in the NAcc at the group level was caused by some participants showing poor learning of expected value, as the correlation between Q in NAcc and the different measures of performance (monetary gains, effective choices, and adaptive switches) was not significant (p>0.25).

In order to assess how well the RW model and the Bayesian observer model generated predictions of the BOLD signal, we estimated two comparable GLMs including only R and Q as generated by each model as parametric modulators at the time of the outcome. We then compared the residuals of the respective GLMs on specific ROIs. The RW model generated better predictions of the BOLD signal in NAcc (paired t-test comparing residuals of the respective GLM models within

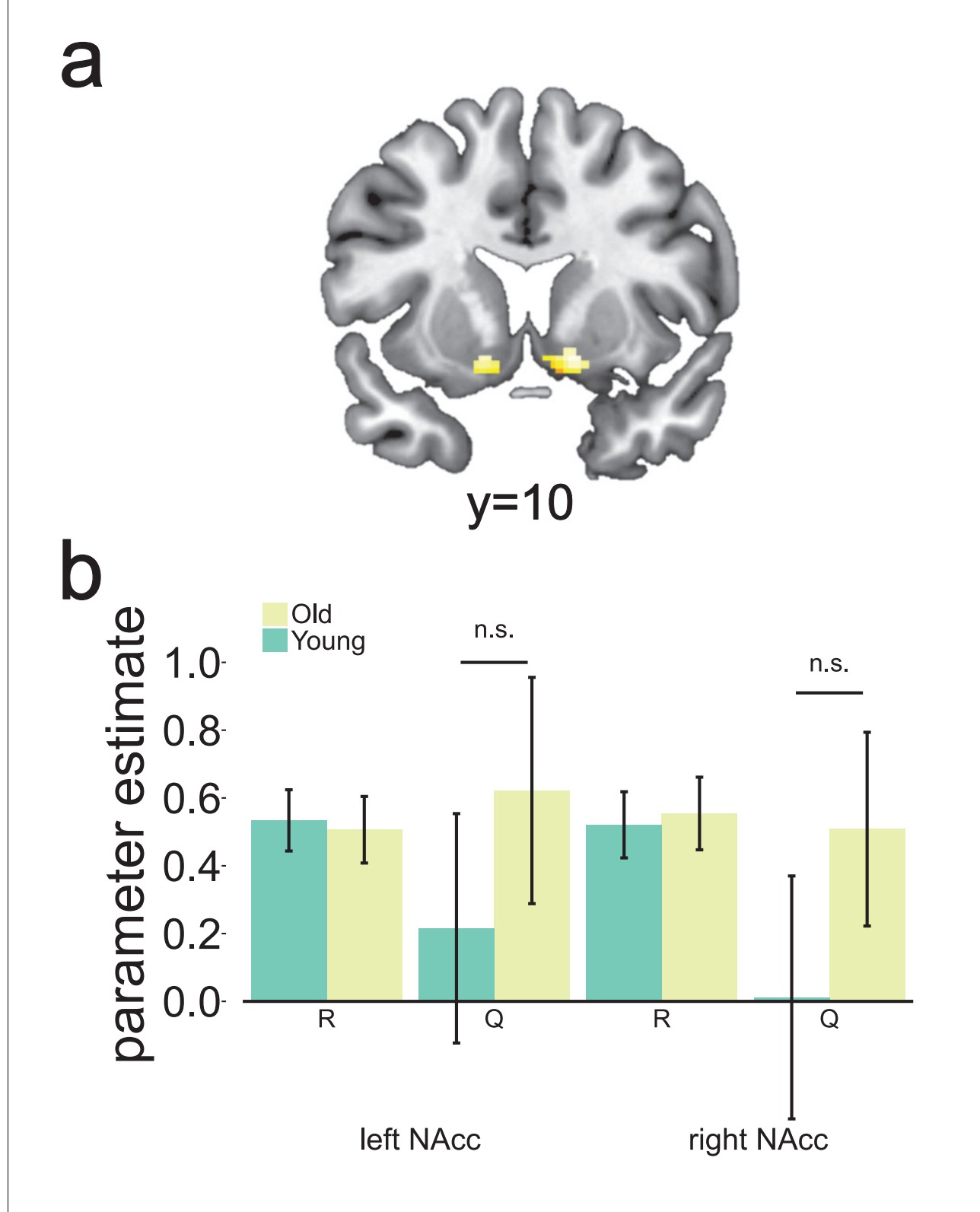

**Figure 4.** Clusters in bilateral NAcc linked to putative reward prediction error (RPE) at the time of the outcome. These were selected as candidate regions to test for canonical RPE showing both a positive effect of reward and a negative effect of Q as calculated by the Bayesian observer model. Extracted parameter estimates for R and Q as calculated by the Bayesian observer model from the regions shown in *Figure 4a*. Although we found a strong effect of reward bilaterally, no expected-value signal was observed for either age group (p>0.10).

*Figure 4 continued on next page*

*Figure 4 continued*

DOI: https://doi.org/10.7554/eLife.26424.012

The following source data and figure supplements are available for figure 4:

**Source data 1.** Activation cluster in ventral striatum as defined by the winning Bayesian model, as well as parameter estimates of R and Q in left and right ventral striatum.

DOI: https://doi.org/10.7554/eLife.26424.015

**Figure supplement 1.** Canonical RPE parameter estimates from the Rescorla-Wagner model.

DOI: https://doi.org/10.7554/eLife.26424.013

**Figure supplement 1—source data 1.** Activation cluster in ventral striatum as defined by the winning Rescorla-Wagner model, as well as parameter estimates of R and Q in left and right ventral striatum.

DOI: https://doi.org/10.7554/eLife.26424.014

functional ROIs; $t(56) = 5.69$, $p<0.001$). This is in line with the extent of the literature showing putative or canonical RPEs as being encoded in NAcc (*Daw et al., 2011*; *McClure et al., 2003*; *O''Doherty et al., 2003*), because the RW model used RPEs to learn the value of actions. On the other hand, the Bayesian observer model generates better predictions of the BOLD signal in the vmPFC when Q as generated by each model was included as a parametric modulator at the time of choice (paired t-test comparing residuals of the respective GLM models across all voxels in the respective vmPFC ROIs; $t(56) = 5.62$, $p<0.001$).

## Relationship to D1 DA BP

We also investigated the relationship among DA D1 BP, age, brain function and performance. We collected PET data using [$^{11}$C]SCH23390 radiotracer that allows DA D1 BP to be measured across the whole brain. We calculated D1 BP in seven a priori ROIs. The selected ROIs were dlPFC, vlPFC, OFC, and vmPFC in cortex, and putamen, caudate and NAcc in striatum in each hemisphere. BP values were calculated and averaged across hemispheres. The selected regions were chosen based on their relevance to our task, as they have previously been reported to be important for various cognitive processes, ranging from value learning and reward sensitivity to working memory and cognitive flexibility (see SI for details). Younger participants had higher values for binding potentials in all ROIs considered ($p<0.001$ in all seven ROIs, *Figure 1*). BP in none of the ROIs was correlated with any measure of performance or any of the model parameters after controlling for multiple comparisons ($p>0.02$; adjusted threshold when controlling for 42 comparisons: $p=0.001$, *Supplementary file 2a*). D1 BP among ROIs was highly correlated after controlling for age ($r(53) = 0.411–0.911$, $p<0.001$, *Supplementary file 2b*).

The group difference in value signals in PFC could be a result of the well-documented age-related decline in DA availability (*Volkow et al., 1998*; *Bäckman et al., 2010*). To investigate this, we performed linear regressions predicting the strength of the link between Q and BOLD in vmPFC from DA D1 BP in all PET ROIs. Because of the high correlation between age and BP in all ROIs ($r(56) > 0.73$, $p<0.001$), we first examined the relationship between BP and Q in vmPFC without controlling for age. BP in NAcc and putamen were related to Q in vmPFC after correcting for multiple comparisons (corrected threshold considering seven ROIs $p=0.007$; NAcc: $r(56) = 0.41$, $p=0.002$; putamen: $r(56) = 0.36$, $p=0.006$). When controlling for age as a predictor of no interest, this correlation only survived for NAcc ($r(53) = 0.28$ $p=0.038$, *Figure 3e*). This result was confirmed by a mediation analysis: Age was a significant predictor of both BP in NAcc ($r(54) = -0.78$, $p<0.001$) and Q in vmPFC ($r(55) = -0.32$, $p=0.016$). BP in Nacc was also a significant predictor of Q in vmPFC $r(54) =0.41$, $p=0.001$. Age was no longer a significant predictor of Q in vmPFC after controlling for BP in NAcc (beta age $= −0.01$, $p=0.964$; beta BP in NAcc $= 0.42$, $p=0.038$). This is consistent with a full mediation of age effects on Q in vmPFC by DA D1 BP in NAcc. Further, despite the main effect of age on D1 BP in NAcc, there was no significant interaction between age group and NAcc D1 BP ($F(1,52) = 1.20$; $p=0.279$) in modelling Q in vmPFC; thus, the relationship between DA D1 BP in NAcc and Q in vmPFC did not differ between age groups.

We did not find any significant relationship between the representation of Q in NAcc at outcome time and D1 BP in any of the ROIs examined ($p>0.11$ in bivariate correlations; $p>0.13$ when controlling for age).

## Discussion

We used a probabilistic reward learning task along with computational modelling, PET measures of the D1 system and fMRI in healthy, cognitively high functioning younger and older participants to investigate the effects of age on value-based decision making and its modulation by DA. We showed that probabilistic reward learning was impaired in older compared to younger participants. We also showed that value anticipation in vmPFC predicted performance beyond age and was attenuated in older participants. Furthermore, the value signal in vmPFC was modulated by D1 BP in NAcc. Finally, our computational model showed that the tendency for choice perseveration can be described as aversion to the variance of the unchosen option and that, for most participants, greater subjective confidence in a previous choice promoted switches away from that choice.

### Dopamine, aging and value signals

An age-related impairment in probabilistic reward learning has been widely reported (*Mell et al., 2005*; *Eppinger et al., 2013*; *Chowdhury et al., 2013*; *Eppinger et al., 2015*; *Samanez-Larkin et al., 2012*). The age-related deterioration of the dopaminergic system (*Volkow et al., 1998*) has been hypothesised to underlie age-related cognitive decline (*Volkow et al., 1998*; *Bäckman et al., 2010*). One mechanism through which DA deficits can affect probabilistic learning performance in aging is by attenuation of value signals in the brain (*Halfmann et al., 2016*). Anticipatory value signals are commonly reported in vmPFC (*Rolls et al., 2008*; *Kim et al., 2011*) as well as in striatum (*Schönberg et al., 2007*; *Behrens et al., 2008*) and are modulated by DA (*Pessiglione et al., 2006*; *Chowdhury et al., 2013*; *Schlagenhauf et al., 2013*). Additionally, RPEs detected in NAcc (*Wimmer et al., 2014*; *Samanez-Larkin et al., 2012*; *Kim et al., 2011*) are thought to reflect dopaminergic signals from midbrain (*Bayer and Glimcher, 2005*; *Schultz et al., 1997*), supporting optimal action selection in probabilistic reward learning (*Frank et al., 2004*).

We found a robust value anticipation signal in vmPFC in both age groups, which is in keeping with neuroimaging findings across a range of similar tasks (*Daw et al., 2006*; *Wimmer et al., 2014*). As expected, this signal was attenuated in the older compared with the younger sample. Furthermore, the strength of the signal predicted performance on the task beyond age and was related to D1 BP in NAcc. Our results are consistent with a full mediation of the age effects on performance by Q in vmPFC, that is, age no longer predicts performance when controlling for the strength of BOLD that reflects Q in vmPFC. The same is true for the strength of Q in vmPFC: the effect of age can be explained by lower DA D1 BP in the older age group. Note, however, that it is difficult to make inferences on mediation effects of age in a cross-sectional dataset (*Lindenberger et al., 2011*). To the best of our knowledge, this is a novel finding demonstrating a relationship between integrity of the mesolimbic DA system and the prefrontal value signal supporting probabilistic learning in humans. This suggests that age-related deficits in probabilistic learning may reflect DA decline blurring value anticipation in vmPFC.

It is unsurprising that anticipatory value signals have a great impact on the ability to perform the present task, considering that damage to vmPFC/medial orbitofrontal cortex (mOFC) in humans and monkeys impairs value-guided decision making (*Halfmann et al., 2016*; *Camille et al., 2011*; *Noonan et al., 2010*; *Rudebeck and Murray, 2014*; *Rushworth et al., 2011*). The nature of this signal is still debated (*Noonan et al., 2012*), as is the cross-species generalizability for prefrontal regions (*Neubert et al., 2015*). Some have proposed that vmPFC tracks the value of items regardless of their nature, because vmPFC activation reflects the value across a range of tasks with different reward features from money to aesthetic and social rewards (*Behrens et al., 2008*; *Kim et al., 2011*; *McNamee et al., 2013*; *O'Doherty, 2007*; *Philiastides et al., 2010*). Others have proposed that vmPFC performs value comparisons, because neural signals represent the value difference between alternative options (*Rushworth et al., 2011*; *Boorman et al., 2009*; *Chau et al., 2014*). Regardless of its exact nature, our findings show that the signal is important not only for reward learning in general but that its attenuation is linked to age-related deficits in probabilistic learning. This notion fits with previous suggestions that age-related impairment in probabilistic learning relates to deficits in PFC function (*Hämmerer and Eppinger, 2012*; *Samanez-Larkin and Knutson, 2015*). Our results show that performance in the TAB is supported by the expected value signal in the vmPFC and that the strength of this signal explains the effects of age on performance. However, considering that the TAB can be seen as noisy reversal learning task, it is a possibility that differences in executive

functions - such as the ability to inhibit a response to previously rewarded option - contribute to group differences in our task (*Bari and Robbins, 2013*).

Value anticipation in vmPFC was modulated by D1 BP in NAcc across both age groups and when controlling for age, again showing a full mediation of age effects on vmPFC signals by DA in NAcc. This finding is in agreement with the view that gating and selection of relevant information in cortex relies on processing within corticostriatal loops (*Shipp, 2017*), which is modulated by DA (*Reynolds and Wickens, 2002*). Pharmacological evidence in humans suggests that D2 receptors have a role in modulating gating of information in working memory (*Cools and D'Esposito, 2011*), but experiments studying this process with selective pharmacological manipulations of the D1 system in humans are lacking. However, computational work suggests a role for striatal D1 receptors in cortico-striatal gating (*Gruber et al., 2006*). The value representation in vmPFC might therefore emerge through this DA-modulated iterative gating process in NAcc. Although BPs are highly correlated across ROIs, a mediation analysis was only significant for the NAcc. This is compatible with the literature on reward processing in the corticostriatal loops. The critical nodes for processing of reward information and motivation are NAcc and the mOFC, including vmPFC (*Haber and Knutson, 2010*). Our data suggest that good performance, based on selection of adaptive actions, relies on D1 availability in NAcc, which in turn allows for robust value anticipation in vmPFC. Note, however, that the relationship between D1 BP and performance was not significant when controlling for age, which precludes inferences about a direct role of DA on performance.

Aside from considering expected value in vmPFC, one might have hypothesised that attenuated RPEs in NAcc of older participants would account for the age-related performance deficit (*Chowdhury et al., 2013*), because of the connection between DA and RPEs in NAcc. This hypothesis builds on the common observation that RPE signals in NAcc are present in younger adults (*McClure et al., 2003*; *O''Doherty et al., 2003*). In contrast to this, we did not observe neural activity reflecting a canonical RPE signal in NAcc in either age group. Although we found a significant effect of reward, we did not obtain a negative effect of expected value. Note that we did not find a canonical RPE in NAcc when using the best of the RW models either. This suggests that the lack of expected value signal in NAcc is not merely caused by generating expected value with the Bayesian ideal observer model which does not make use of RPEs to update value representations.

The lack of canonical RPE signal in NAcc could stem from the fact that we used a very stringent test for RPEs. Previous studies using the same stringent method report mixed results. Whereas some studies report significant positive effects of reward obtainment and negative effects of expected value (*Behrens et al., 2008*; *Niv et al., 2012*), others do not find this canonical signal in NAcc (*Chowdhury et al., 2013*; *Stenner et al., 2015*; *Wimmer et al., 2014*; *Li and Daw, 2011*). The conditions under which a canonical RPE can be detected may depend on task characteristics. For example, if the RPE signal is not behaviourally relevant for the task at hand it may not be encoded in the NAcc. In our case, however, RPEs are behaviourally relevant because the choice between bandits is based on fine-grained differences in their values. However, for other paradigms, the lack of behavioural relevance of RPEs could potentially explain a negative result (*Stenner et al., 2015*; *Guitart-Masip et al., 2012*). Another important aspect may be the temporal proximity of the choice cues and the outcome presentation in the task. This may hinder the dissection of opposing responses to these events with fMRI. We cannot rule out the possibility that our negative result stems from this feature of our task design and for this reason, we cannot provide conclusive evidence on the lack of canonical RPE signal in the NAcc. Our results point, however, to the need for stringent tests in future studies of the neural underpinnings of RPEs with fMRI.

The lack of canonical RPEs in older participants has already been observed using the same task (*Chowdhury et al., 2013*). In that study, canonical RPEs were detected in NAcc of older participants after boosting the dopaminergic system with levodopa. These findings were interpreted as evidence that older participants had deficient RPEs signals in NAcc due to DA decline, and that remediating this deficit could restore the RPE signal. However, no younger comparison group was scanned to confirm that the deficient expected value signal observed in the older participants on placebo was age-specific. *Chowdhury et al. (2013)*, nevertheless, showed that the expected value signal in NAcc is sensitive to DA manipulations. Contrary to what one might expect from these data, the relationship between expected value (Q) as predicted by the winning model and NAcc BOLD signal was not modulated by D1 BP in any ROI considered. The reason for this negative result remains unknown. In striatum, D1 receptors have lower affinity to DA than D2 receptors and their stimulation is

hypothesised to be dependent on phasic changes in DA (*Maia and Frank, 2011*). Because RPE in NAcc is thought to reflect phasic fluctuations of DA levels (*Schultz et al., 1997*), one would expect that D1 receptors would be sensitive to these fluctuations. Our results do not support this view. An alternative account is that the dopaminergic modulation of BOLD signal in NAcc observed by *Chowdhury et al. (2013)* after administration of levodopa is related to stimulation of D2 rather than D1 receptors. Supporting this view, recent evidence suggests that D2 receptors can encode both tonic and phasic DA signals in striatum (*Marcott et al., 2014*).

## Computational mechanisms of switch behaviour

Using computational modelling, we explored different possible influences on the trade-off between exploration and exploitation in the probabilistic reward-learning task. We considered two families of computational models, variations of a standard RL model using RPEs to learn the mean expected value of the bandits and variations in Bayesian ideal observer model that tracked the probability of obtaining a reward for each bandit as a beta distribution. In both model families, including a parameter that promoted forgetting of the unchosen bandit improved model fit. Similarly, including a perseveration parameter to account for the tendency to repeat choices regardless of expected value (*Rutledge et al., 2009*; *Schönberg et al., 2007*; *Lau and Glimcher, 2005*) improved model fit in both families. However, a Bayesian model that modulated the expected value of the unchosen option by the variance of that option outperformed any model with perseveration. Across participants, the variance of the unchosen option had a negative impact on the value of that option. This is opposite to an exploration bonus or uncertainty based exploration term that arises in various more or less normative accounts of exploration (*Dayan and Sejnowski, 1996*) and has been observed in some experiments (*Badre et al., 2012*; *Wilson et al., 2014*). However, many previous studies of decision-making have also shown that variance may be penalised as a form of risk sensitivity (*Symmonds et al., 2011*; *Payzan-LeNestour et al., 2011*; *d'Acremont et al., 2013*), and this is a cousin of the effect that we observed. Furthermore, our model comparison showed that uncertainty aversion is a better account of the perseveration typically observed in bandit tasks (*Rutledge et al., 2009*; *Schönberg et al., 2007*) than a choice kernel. This is a novel insight into the mechanism usually referred to as perseveration and suggests that aversion to the uncertainty about the option that was not chosen previously causes a tendency to stick to ones choices. Whether perseveration observed in other paradigms can be accounted for in the same way remains unknown.

Additionally, a Bayesian model that modulated the value of the most recent choice by the relative subjective confidence in that choice outperformed all other models. Increased relative confidence about the most recent choice resulted in increased attractiveness for the other option. This implies that participants were more likely to switch away from the most recent choices as their subjective confidence in those choices increased. This may appear counterintuitive, as one would expect that increased confidence would lead to choice repetition (*Vinckier et al., 2016*). However, performance improved as the effect of relative confidence decreased, and those participants showing the highest performance had the reverse effect of confidence on choice. In other words, these participants' behaviour was consistent with a negative confidence parameter rather than a positive confidence parameter, implying that increased confidence in previous choices promoted staying with the previously chosen option. One reason for the unwarranted use of confidence in the majority of participants could stem from participants perceiving the task as highly volatile. As a result, they may infer that increasing confidence in the most recent choice indicates that the unchosen option has become better than the chosen option (*Behrens et al., 2007*; *Mathys et al., 2011*). Additionally, the observed effect of κ could reflect safe exploration: if the participant is convinced they have recently chosen the best option a lot (hence their confidence), they can afford to explore the more uncertain option. These possibilities provide interesting directions for future research.

Despite the performance difference, we did not find age differences in any single model parameter, precluding any conclusions about which computational process is affected in old age. In fact, it is likely that the process underlying age differences in performance is not parametrised in the winning Bayesian model. This stands in contrast with the less accurate but simpler RW model, in which the effect of aging was consistently manifested in the learning rate (*Supplementary file 1c*).

## Conclusions

We measured brain activity in younger and older adults performing a probabilistic learning task and found that a signal in vmPFC at the time of choice reflecting expected value was correlated to successful performance. This activity was dependent on DA availability and age, providing support for age-related prefrontal and dopaminergic alterations as candidate mechanisms for impaired probabilistic reward learning and subsequent optimal action selection commonly reported in aging. These results provide insights into the neural and behavioural underpinnings of probabilistic learning and highlight the mechanisms by which age-related dopaminergic deterioration impacts decision making.

# Materials and methods

## Participants

Thirty healthy older adults aged 66–75 and thirty younger adults aged 19–32 were recruited through local newspaper advertisements in Umeå, Sweden. Sample size and power were calculated based on previous studies. One was a study of DA D1 BP differences between age groups (*Rieckmann et al., 2011*). The authors found clear differences in DA D1 BP after testing 20 participants in each age group: Cohen's d = 3.00 (pooled SD = 0.04) for frontal and parietal areas, Cohen's d = 1.60 (pooled SD = 0.21) for striatal ROIs. Assuming this difference, in order to obtain 90% power on a two-tailed independent sample t-test, 10 participants were needed in each age group. Additionally, to estimate the appropriate sample size for the behavioural task, we used the previous study by *Chowdhury et al. (2013)*, who found a behavioural difference on the same task between younger and older participants of similar age ranges: Cohen's d = 0.57 (pooled SD = 0.99). Assuming this difference, in order to obtain 70% power on a one-tailed t-test of a behavioural difference between two samples, 30 participants were needed in each group. Higher power could not be reached, due to financial constraints posed by the cost of PET scans.

The health of all potential participants was assessed before recruitment by a questionnaire administered by the research nurses. The questionnaire enquired about past and present neurologic or psychiatric conditions, head trauma, diabetes mellitus, arterial hypertension that required more than two medications, addiction to alcohol or other drugs, and bad eyesight. All participants were right-handed and provided written informed consent prior to commencing the study. Ethical approval was obtained from the Regional Ethical Review Board. Participants were paid 2000 SEK (~$225) for participation and earned up to 149 additional SEK (~$17) in the two-armed bandit task (TAB). Three older participants were excluded from the TAB analysis, one due to excessive head motion during fMRI scanning, one for only ever selecting one of the two stimuli in the task, and one due to a malfunctioning button box, resulting in no recorded responses. One additional older participant did not complete the full PET scan, but this participant's fMRI and task data are still included in the analysis where possible. This resulted in a total of 57 participants for fMRI and task analysis (27 old, 30 young) and 56 participants for PET analysis (26 old, 30 young).

## Procedure

Participants completed a health questionnaire via telephone prior to recruitment. All participants performed the Mini Mental State Examination (MMSE). Scores ranged from 26 to 30 in the young sample (mean = 29.4, SD = 0.97) and from 27 to 30 in the older sample (mean = 29.4, SD = 0.77), with no difference between the two (p=0.89). PET and fMRI scanning were planned 2 days apart. However, due to a technical problem with the PET scanner, 12 participants were scanned at a longer delay apart (range 4–44 days apart). On the MRI scanning day, participants completed the TAB and another unrelated task inside the MRI scanner. Participants also completed a battery of tasks outside the scanner. Only results from the TAB will be discussed here.

## Two-armed bandit task

The TAB (10) was presented in Cogent 2000 (Wellcome Trust for Neuroimaging, London, UK). *Figure 1a* depicts a schematic representation of one TAB trial. Participants were instructed to choose the fractal stimulus they thought to be most rewarding at each trial and were informed of the changing probability of obtaining a reward for each stimulus. These probabilities varied

independently from one another. Probabilities were generated using a random Gaussian walk (*Daw et al., 2006*). Before scanning, participants were presented with five practice trials. The same set of random Gaussian walks was used for all participants, but assignment of random walk to stimulus identity was counterbalanced across participants.

## Computational modelling of behavioural data

We built a variety of different models which can be classified into two main families. The first includes variations on standard RL models whereby action values are learned through reward prediction errors (RPEs) using the RW updating rule. The second family of models include variations on a Bayesian ideal observer whereby the probability distribution of obtaining a reward is updated after each outcome observation. All models, regardless of family, use a softmax rule with an inverse temperature parameter $\beta$ (with $\beta > 0$) to determine the probability that the participants chooses action a:

$$P(a(t) = a) = \frac{\exp[\beta m_a(t)]}{\exp[\beta m_0(t)] + \exp[\beta m_1(t)]} \tag{1}$$

here, $m_a(t)$ is the propensity for selecting action a. The next section lays out how $m_a(t)$ is defined in the models we explored.

Reinforcement learning models

For RL models, expected values (Q) for trial t were calculated for each action $a \in \{0,1\}$ (corresponding to each bandit). $Q_a(t + 1)$ is calculated according to standard RW updating rule:

$$Q_{a(t)}(t+1) = Q_{a(t)}(t) + \alpha \delta(t) \tag{2}$$

$$\delta(t) = R(t) - Q_a(t)(t) \tag{3}$$

$Q_{a(t)}(t)$ is the expected value of the option $a(t)$ selected on trial t. Q for both actions was set to 0.5 at the start of the experiment. $\delta(t)$ is the difference between expected value and received reward (R) on trial t. R is a binary with the value of 1 on rewarded trials, and 0 on unrewarded trials. $\alpha$ is the learning rate, with $0 < \alpha < 1$, indicating the weight given to the RPE on the current trial. A greater value for $\alpha$ results in faster updating of Q.

In the simplest model, $m_a(t) = Q_a(t)$. We included an additional parameter in the definition of $m_a(t)$: a perseveration parameter b (with $-\infty < b < \infty$), reflecting the common observation that participants tend to either repeat their choices, or avoid repetition (*Rutledge et al., 2009*; *Schönberg et al., 2007*; *Lau and Glimcher, 2005*). This parameter raises or lowers the expected value of a stimulus if that stimulus was also chosen on the previous trial. Thus,

$$m_a(t) = Q_a(t) + b\chi_{a=a(t-1)} \tag{4}$$

where a positive value of b reflects a tendency to perseverate (repeat the same choice), and a negative value reflects avoiding perseveration.

We considered another definition of $m_a(t)$, where in addition to the perseveration parameter b, we considered the possibility that the unchosen stimulus may decay in value each time it is not selected by the participant. This was instantiated by the inclusion of a 'forget' parameter $\varphi$ (with $0 < \varphi < 1$)(*Barch et al., 2003*), so that the Q value for the unchosen option relaxes towards 0.5. Thus,

$$Q_a(t+1) = Q_a(t) + \varphi(0.5 - Q_a(t))\chi_{a \neq a(t1)} \tag{5}$$

In this model, the value of the chosen option is updated as described in *Equation 2*.

Bayesian observer models

Choice behaviour was modelled by representing the probability of obtaining a reward for each possible action $a \in \{0, 1\}$ (corresponding to each bandit) as a beta distribution

$$\theta_a \sim \beta(\theta_a; \gamma_a, \varepsilon_a) \tag{6}$$

that is updated upon observation of the outcome on each trial. On any given trial, these models

generate expectations about the mean probability of obtaining a reward (which we will refer to as $Q_a(t)$, for consistency with the RL models) and its variance ($V_a(t)$):

$$Q_a(t) = \frac{\gamma_a}{(\gamma_a + \varepsilon_a)} \tag{7}$$

$$V_a(t) = \frac{\gamma_a \varepsilon_a}{(\gamma_a + \varepsilon_a)^2 (\gamma_a + \varepsilon_a + 1)} \tag{8}$$

The parameters of the beta distributions were initialised at 1 ($\gamma_a = \varepsilon_a = 1$). This implies that $Q_0(1) = Q_1(1) = 0.5$ and maximum variance $V_0(1) = V_1(1) = 0.143$ reflecting an expectation of reward equal to chance for both bandits and a lack of knowledge about the underlying probability distributions. After getting a reward for choosing action $a$, $\gamma_a$ is increased by 1 and both $\gamma_a$ and $\varepsilon_a$ are relaxed towards 1. Conversely, after reward omission, $\varepsilon_a$ is increased by 1 and both $\gamma_a$ and $\varepsilon_a$ are relaxed towards 1. Hence,

$$\begin{aligned}\gamma_{a(t)}(t+1) &= (1-\omega)\gamma_{a(t)}(t) + \omega + 1; &&\text{and}\\ \varepsilon_{a(t)}(t+1) &= (1-\omega)\varepsilon_{a(t)}(t) + \omega; &&\text{if } R(t) = 1\end{aligned} \tag{9}$$

$$\begin{aligned}\gamma_{a(t)}(t+1) &= (1-\omega)\gamma_{a(t)}(t) + \omega; &&\text{and}\\ \varepsilon_{a(t)}(t+1) &= (1-\omega)\varepsilon_{a(t)}(t) + \omega + 1; &&\text{if } R(t) = 0\end{aligned} \tag{10}$$

For the unchosen bandit, both $\gamma_a$ and $\varepsilon_a$ are relaxed towards 1:

$$\begin{aligned}\gamma_{1-a(t)}(t+1) &= (1-\lambda)\gamma_{1-a(t)}(t) + \lambda; &&\text{and}\\ \varepsilon_{1-a(t)}(t+1) &= (1-\lambda)\varepsilon_{1-a(t)}(t) + \lambda;\end{aligned} \tag{11}$$

$\omega$ and $\lambda$ are individual participants' freeparameters governing how fast reward probabilities are updated ($\omega$, with $0 < \omega < 1$) and forgotten ($\lambda$, with $0 < \lambda < 1$). In the simplest model we considered, $\omega = \lambda$. We also considered the possibility that updating and forgetting mechanisms occurred at different speeds, hence allowing $\omega$ and $\lambda$ to be different.

As stated previously, $m_a(t)$ reflects the propensity of selecting action $a$, where the simplest definition of $m_a(t)$ is $m_a(t) = Q_a(t)$ as defined in *Equation 7*, either calculated from a model with one single update parameter ($\omega = \lambda$) or with two separate update parameters ($\omega \neq \lambda$).

We then considered a variety of possible additions to $m_a(t)$ which reflected various factors that might influence choice. We tested different combinations of nested models using methods of model comparison. First was choice perseveration $b\chi_{a=a(t-1)}$ just as in *Equation 4*.

The second potential addition concerned the variance $V_a(t)$ of the beta distributions for the individual bandits. In principle, since the subjects might have framed their decision as being between sticking and switching, there could be separate influences associated with the bandit that was or was not chosen on the previous trial. Thus, we considered two separate contributions:

$$\upsilon^{\text{chosen}} V_a(t) \chi_{a=a(t-1)} \quad \text{and} \tag{12}$$

$$\upsilon^{\text{unchosen}} V_a(t) \chi_{a=1-a(t-1)}. \tag{13}$$

If $\upsilon^{\text{chosen}}$ or $\upsilon^{\text{unchosen}}$ are positive, then there is a tendency to choose in favour of high variance – a form of uncertainty or exploration bonus.

Finally, we considered the possibility that subjective confidence that participants can calculate about the correctness of their choices might modulate choice. Based on *Sanders et al. (2016)*, confidence (C) can be defined as:

$$C = P(\text{correct}|\text{observations, choice}) \tag{14}$$

Given that our Bayesian observer model tracks subjective estimates of the mean and the variance of the probability distribution of obtaining a reward for each bandit, the probability in *Equation 14* can be approximated by:

$$\mathrm{C}_1(\mathrm{t}) = \mathrm{P}(\theta_1 > \theta_0) = \int_{\theta_1=0}^{1} \mathrm{d}\theta_1 \beta(\theta_1;\, \gamma_1,\, \varepsilon_1) \int_{\theta_0=0}^{\theta_1} \mathrm{d}\theta_0 \beta(\theta_0;\, \gamma_0,\, \varepsilon_0) \tag{15}$$

$$\mathrm{C}_0(\mathrm{t}) = \mathrm{P}(\theta_0 > \theta_1) = 1 - \mathrm{C}_1(\mathrm{t}) \tag{16}$$

Given the simple relationship between these two confidences, there are various essentially equivalent ways of incorporating it into choice. We considered the relative confidence in the choice on a trial:

$$\mathrm{C}^{\mathrm{rel}}(\mathrm{t}) = \mathrm{P}\left(\theta_{\mathrm{a(t)}} > \theta_{1-\mathrm{a(t)}}\right) - \mathrm{P}\left(\theta_{1-\mathrm{a(t)}} > \theta_{\mathrm{a(t)}}\right) = 2\mathrm{P}\left(\theta_{\mathrm{a(t)}} > \theta_{1-\mathrm{a(t)}}\right) - 1 \tag{17}$$

and assessed the extent to which the relative confidence on trial $\mathrm{t}-1$ encouraged switching on trial $\mathrm{t}$ by adding a factor $\kappa \mathrm{C}^{\mathrm{rel}}(\mathrm{t}-1)\chi_{\mathrm{a}=1-\mathrm{a(t}-1)}$ to the action that was not chosen on trial $\mathrm{t}-1$. Here, positive values of $\kappa$ make the subjects more likely to switch if they had been more confident.

## Model fitting and comparison

Model parameters were fitted using an expectation-maximisation approach (*Guitart-Masip et al., 2012*; *Huys et al., 2011*). We used a Laplacian approximation to obtain maximum a posteriori estimates for the parameters for each participant iteratively, starting with flat priors. After an iteration, the resulting group mean posterior and variance for each parameter were used as priors in the next iteration. This method was used to prevent the individuals' parameters from taking on extreme values.

Models were compared using the integrated Bayesian Information Criterion (iBIC) (*Guitart-Masip et al., 2012*; *Huys et al., 2011*), where small iBIC values indicate a model that fits the data better after penalizing for the number of parameters. Comparing iBIC values is akin to a likelihood ratio test.

## Statistical analysis of behaviour and brain variables

We calculated a number of behavioural measures: (1) the total monetary gains in Swedish Crowns (SEK), (2) percentage of efficient choices (the proportion of choices in which participants chose the option that was most likely to be rewarded according to the random Gaussian walks), (3) number of switches between bandits, and (4) percentage of adaptive switches, defined as switches to subjectively better bandits (according to the winning model) versus switches to subjectively worse bandits. We used independent sample one-tailed t-tests to assess group differences in task performance, based on previously reported observations of impaired probabilistic reward learning performance in old age (*Eppinger et al., 2011*; *Mell et al., 2005*). We hypothesised that the older group mean would be lower than the young group mean. Non-parametric independent two-tailed two sample Mann-Whitney tests were used to assess group differences in model parameters and other variables that were non-normally distributed. Regular two-tailed two-sample t-tests were used elsewhere. Pearson's correlations were used to analyse the data further, controlling for age and model fit, as defined by the participant's log likelihood, where appropriate. Statistical analyses were performed in SPSS 22 and R3.3.0.

## MRI acquisition

Brain imaging data were acquired on a 3.0TE MR-scanner (GE Medical Systems). T1-weighted 3D-SPGR images were acquired using a single-echo sequence (voxel size: $0.5 \times 0.5 \times 1$ mm, TE = 3.20, flip angle = 12 deg). Functional images were acquired using a T2*-sensitive gradient echo sequence (voxel size: $2 \times 2 \times 4$ mm, TE = 30.0 mis, TR = 2000 ms, flip angle = 80 deg), and contained 37 slices of 3.4 mm thickness, with a 0.5 mm gap between slices. Volume acquisition occurred in an interleaved fashion. 330 volumes were obtained for each of the two functional runs. During acquisition of fMRI time series, heart rate and respiratory data were collected using a breathing belt and a pulse oximeter.

## MR analysis

fMRI analysis was performed using SPM8 (http://www.fil.ion.ucl.ac.uk/spm/software/spm8/). Preprocessing steps included (in this order): slice-time correction, realignment, coregistration to the T1-weighted image, movement correction using ArtRepair (see below), normalisation to MNI space using a diffeomorphic registration algorithm (DARTEL) as implemented in SPM (*Ashburner, 2007*) with spatial resolution after normalisation 2 × 2×2 mm. Data were smoothed with a final Gaussian kernel equivalent to a standard 8 mm. This kernel was achieved in two steps, including the ArtRepair motion correction (see below). The fMRI time series data were high-pass filtered with cut-off 128 s, and whitened with an AR(1) model. For each participant, the canonical hemodynamic response function was used to compute their statistical model.

The movement parameters showed that 15 participants moved >3 mm in any direction during functional runs. To correct for movement artefacts, we used the ArtRepair toolbox (*Mazaika et al., 2005*; *Levy and Wagner, 2011*). ArtRepair assesses the amount of motion between volume acquisitions from the mean intensity plot and linearly interpolates scans in which motion over a user-specified threshold is present. We set our threshold to the recommended value of 1.5% deviation of the mean intensity between scans. The average number of interpolated scans for our participants was 12.2 (1.8%) (SD = 19.6 (3.0%)) and one participant was excluded for showing movement >1.0 mm in >25% of scans, in line with default recommendations. ArtRepair requires smoothing of the individual subject data with a Gaussian smoothing kernel of 4 mm. A Gaussian kernel of 7 mm was then used after the normalisation to MNI space, resulting in a smoothed, normalised image equivalent to a more standard 8 mm smoothed normalised image.

We estimated three first-level models, in order to address the different goals of the study. GLM 1 was set to study how value anticipation is represented in the brain and how this representation differs between age groups and relate to task performance and DA D1 receptor density. GLM 2 and 3 were set to investigate the differences in the expression of the RPE signal at the time of the outcome in the old and young sample and its relation to task performance and DA D1 receptor density as measured by PET. Note that our winning computational model does not use RPEs. However, because our Bayesian observer model generates value expectations, we may expect the brain to, nonetheless, track RPEs as the discrepancy between observed outcomes and outcomes predicted by the model. All GLMs (described in detail below) included a regressor specifying the time of choice and one specifying the time of outcome. These were parametrically modulated by various regressors that were calculated based on the winning computational model and the group posterior parameter means. These regressors are mean-centered by default (*Mumford et al., 2015*). The SPM motion regressors were also added to the design matrix as regressors of no interest, as well as 18 parameters correcting for physiological noise as recorded by a heartbeat detector and breathing belt during the scanning sessions. These were calculated using the PhysiO toolbox version r671 (https://www.tnu.ethz.ch/en/software/tapas.html).

GLM 1: Because the choice and outcome are close in time in each trial (maximum 3 s apart), including Q as a parametric modulator at both time points would result in highly correlated regressors. Therefore, to investigate brain activity reflecting value anticipation, we estimated a model that included Q at the time of the choice. R was included at the time of the outcome as a regressor of no interest. For each participant, we calculated a contrast image weighting the parametric modulators of interest (Q at choice) by 1. At the second level, we used this contrast image to perform a one-sample t-test across age groups. The second-level map was produced with a family-wise error (FWE) corrected threshold at p<0.05 and parameter estimates for Q were extracted from relevant surviving clusters to investigate the relationship between the signal, task performance and DA.

GLM 2 (putative RPE): When investigating RPE signals, a common approach is to identify regions in which activity is correlated with the RPE, defined as $R(t)-Q_a(t)$, included as a single regressor in the GLM (*Eppinger et al., 2013*; *Schönberg et al., 2007*; *McClure et al., 2003*). Because R and RPE are correlated (*Behrens et al., 2008*; *Niv et al., 2012*; *Li and Daw, 2011*), when using this approach the amount of variance attributed to RPE may be overestimated and the identified signals can be seen as putative RPEs. For this reason, it has been suggested that the effects of R and Q need to be estimated separately and only regions showing both signals can be considered as conveying a canonical RPE signal. In order to identify regions potentially conveying a canonical RPE signal, we first identified regions conveying a putative RPE signal, by setting up a first-level GLM including the putative

RPE regressor (R(t)-Q$_a$(t)) as a single parametric modulator at the time of outcome presentation. For each participant, we calculated a contrast image weighting this parametric modulator by 1. At the second level, we used these contrast images to perform a one-sample t-test across age groups. All second-level maps were produced with a family-wise error (FWE) corrected threshold at p<0.05. The bilateral NAcc, commonly reported to respond to RPEs, was identified in this analysis and used as functional ROIs for further analysis. To constrain these ROIs, we used the conjunction of the functional ROIs and the anatomical NAcc masks found in the PickAtlas (https://www.nitrc.org/projects/wfu_pickatlas/).

GLM 3: To quantify the separate RPE components, we performed another first-level analysis in which R and Q were included as two independent parametric modulators at the time of the outcome in the design matrix. For each participant, we calculated a contrast image weighting these two independent parametric modulators by 1. Parameter estimates for R and Q were extracted from these contrast maps using the ROIs defined in the second-level analysis described in GLM 2 and were further analysed to look for a canonical RPE signal.

## Time course extraction

The aim of this analysis was to visualise the effect of variables of interest on the BOLD signal, at the time of the choice and at the time of the outcome. Time courses of BOLD data from specified ROIs were extracted from the preprocessed, normalised EPI images. This BOLD signal was upsampled to one measurement every 200 ms. This time series resampled into chunks of 15 s, corresponding to individual trials. Stimulus onset occurred at 0 s, choice between 0 and 2 s, and outcome at 3 s. A general linear model including the regressors of interest was estimated at each time point in each trial for each participant. In these models, the regressors of interest were allowed to compete for variance. At each time point, group mean effect sizes and standard errors were calculated and plotted separately for young and old.

## PET image acquisition

PET images were acquired in 3D mode using a Discovery 690 PET/CT (General Electric, WI, US), at the Department of Nuclear Medicine, Norrland's University Hospital. A low-dose helical CT scan (20 mA, 120 kV, 0.8 s/revolution), provided data for PET attenuation correction. Participants were injected with a bolus of 200 MBq [11C]SCH 23390. A 55-min dynamic acquisition commenced at time of injection (9 frames x 2 min, 3 frames x 3 min, 3 frames x 4,20 min, 3 frames x 5 min). Attenuation- and decay-corrected 256 × 256 pixel transaxial PET images were reconstructed to a 25 cm field-of-view employing the Sharp IR algorithm (6 iterations, 24 subsets, 3.0 mm Gaussian post filter). Sharp IR is an advanced version of the OSEM method for improving spatial resolution, in which detector system responses are included (*Ross and Stearns, 2010*). The Full- Width Half-Maximum (FWHM) resolution is below 3 mm. The protocol resulted in 47 tomographic slices per time frame, yielding 0.977 × 0.977 × 3.27 mm$^3$ voxels. Images were decay-corrected to the start of the scan. Images were de-identified using dicom2usb (http://dicom-port.com/). To minimise head movement during the imaging session, the patient's head was fixated with an individually fitted thermoplastic mask (Positocasts Thermoplastic; CIVCO medical solutions, IA, USA).

## PET analysis

PET data were analysed in a standard ROI-based protocol. This type of analysis requires a priori hypotheses about the regional specificity of dopaminergic modulation of observed behavioural or neuronal effects. All analyses were done with the use of in-house developed software (imlook4d version 3.5, https://dicom-port.com/product/imlook4d/).

Regions of interest for the ROI analysis were dorsolateral PFC (dlPFC), ventrolateral PFC (vlPFC), orbitofrontal cortex (OFC), and vmPFC in cortex, and putamen, caudate and NAcc in striatum across hemispheres. These regions were chosen based on their relevance to our task: dlPFC has previously been demonstrated to be involved in executive processes and working memory (WM) and cognitive flexibility (*Barch et al., 2003*; *D'Esposito et al., 1995*; *Petrides, 2000*; *Plakke and Romanski, 2016*), whereas vlPFC is thought to be important for goal-directed action and attention (*Levy and Wagner, 2011*). vmPFC has been shown to be responsive to reward magnitude and reward probability in an overwhelming number of studies (*Rushworth et al., 2008*). In addition, vmPFC and OFC

are active during anticipation of rewards (*Kim et al., 2011*). Many connections exist between these regions and ventral striatum (VS), an important node in the mesolimbic dopamine system (*Rushworth et al., 2011*; *Haber and Knutson, 2010*; *Salamone and Correa, 2012*). VS consists of NAcc, and parts of the medial caudate nucleus and rostral putamen. Because of its connections with prefrontal areas relevant to this task, and because striatum is densely innervated by dopaminergic neurons, we segmented the different parts of striatum to use as separate ROIs. The cerebellum was segmented to be used as reference tissue because it is devoid of DA D1 receptors (*Hall et al., 1994*). Freesurfer's recon-all function (*Desikan et al., 2006*) was used to segment the brain into cortical ROIs, FSL's FIRST algorithm (*Patenaude et al., 2011*) was used to segment subcortical structures.

In order to obtain ROI BP values, the PET time series were first coregistered to the individual T1-weighted images and ROI images. The average time activity curves (TAC) were extracted across all voxels within each ROI and calculated binding potential (BP) by applying the Logan method (*Logan et al., 1990*) as implemented in imlook4d. This method was applied to each ROI using the cerebellum as reference tissue. BP values for all ROIs were averaged across hemispheres. We then investigated the relationship between DA D1 BP in the different ROIs and the Q signal in NAcc and vmPFC while controlling for age and model fit.

## Acknowledgements

We thank Mats Erikson and Kajsa Burström for collecting the data.

## Additional information

### Funding

| Funder | Grant reference number | Author |
| --- | --- | --- |
| Vetenskapsrådet | VR521-2013-2589 | Marc Guitart-Masip |
| Gatsby Charitable Foundation | | Peter Dayan |
| Humboldt Research Award | | Lars Bäckman |
| af Jochnick Foundation | | Lars Bäckman |

The funders had no role in study design, data collection and interpretation, or the decision to submit the work for publication.

### Author contributions

Lieke de Boer, Conceptualization, Data curation, Formal analysis, Validation, Investigation, Visualization, Methodology, Writing—original draft, Writing—review and editing; Jan Axelsson, Data curation, Software, Formal analysis, Writing—review and editing; Katrine Riklund, Conceptualization, Resources, Project administration, Writing—review and editing; Lars Nyberg, Conceptualization, Resources, Supervision, Investigation, Methodology, Project administration, Writing—review and editing; Peter Dayan, Data curation, Software, Formal analysis, Investigation, Visualization, Methodology, Writing—review and editing; Lars Bäckman, Conceptualization, Resources, Supervision, Investigation, Methodology, Writing—review and editing; Marc Guitart-Masip, Conceptualization, Resources, Data curation, Software, Formal analysis, Supervision, Funding acquisition, Validation, Investigation, Visualization, Methodology, Writing—original draft, Project administration, Writing—review and editing

### Author ORCIDs

Lieke de Boer (ID) http://orcid.org/0000-0003-3381-2040

Peter Dayan (ID) http://orcid.org/0000-0003-3476-1839

Marc Guitart-Masip (ID) http://orcid.org/0000-0002-2294-6492

## Ethics

Human subjects: Ethical approval was obtained from the Umeå Ethical Review Board, identifier DNR 2014-251-31M. All participants provided written informed consent prior to commencing the study.

## Decision letter and Author response

Decision letter https://doi.org/10.7554/eLife.26424.026
Author response https://doi.org/10.7554/eLife.26424.027

# Additional files

## Supplementary files

• Source code 1. Computational modelling. Scripts needed for the entire modelling routine used in the behavioural analysis. See the comments in the file fit_all_models_eLife.m for more details on each of the models and the procedure
DOI: https://doi.org/10.7554/eLife.26424.016

• Source code 2. fMRI analysis. All MATLAB scripts required to set up preprocessing of fMRI data, create regressors for fMRI analysis, run the first level analysis and the second level analysis.
DOI: https://doi.org/10.7554/eLife.26424.017

• Source code 3. PET analysis. Scripts required to run the segmentation of T1 images, PET analysis and estimation of BPs for the different ROIs.
DOI: https://doi.org/10.7554/eLife.26424.018

• Source code 4. figures. R script for ggplot for *Figures 1b*, *3b, d and e* and *4b*
DOI: https://doi.org/10.7554/eLife.26424.019

• Source code 5. *Figure 2*. MATLAB script that creates joint probability distributions shown in *Figure 2*.
DOI: https://doi.org/10.7554/eLife.26424.020

• Source code 6. timecourse extraction. MATLAB script that extracts the timecourse for expected value from vmPFC for young and old separately.
DOI: https://doi.org/10.7554/eLife.26424.021

• Supplementary file 1. (A) Correlation coefficients between model parameters and performance. Coefficients in italics represent significant correlations at $p<0.05$. Coefficients in bold represent significant correlations at $p<0.002$ (adjust Bonferroni-corrected threshold). (B) Variance in number of switches as explained by the strongest RW model and winning model. When explaining the number of switches from the individual model parameters, the parameters that weighted V ($\upsilon$), $C_{rel}$ ($\kappa$) and forgetting rate ($\lambda$), in addition to the softmax temperature parameter ($\beta$) were found to be significant predictors. Age or other model predictors did not contribute significantly. This regression model explained the number of switches better than the RW model parameters, where only the perseveration parameter b and softmax temperature parameter $\beta$ were significant predictors of number of switches. (C) Young participants have a higher learning rate in the winning Rescorla-Wagner model according to non-parametric t-tests. None of the other model parameters significantly differed between groups.
DOI: https://doi.org/10.7554/eLife.26424.022

• Supplementary file 2. (A) No significant correlations between model parameters and dopamine D1 receptor density in any ROI after controlling for age at Bonferroni-corrected threshold of 0.0014. (B) Partial correlation matrix showing correlation coefficients between the binding potential in the different PET ROIs and their p-values after controlling for age.
DOI: https://doi.org/10.7554/eLife.26424.023

• Supplementary file 3. Coordinates of clusters responsive to Q at the time of choice.
DOI: https://doi.org/10.7554/eLife.26424.024

• Transparent reporting form
DOI: https://doi.org/10.7554/eLife.26424.025

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
