## [Decision Letter]

Thank you for submitting your article "Dopaminergic, neural and computational contributions to probabilistic reward learning in old age" for consideration by *eLife*. Your article has been reviewed by three peer reviewers, and the evaluation has been overseen by a Reviewing Editor and Sabine Kastner as the Senior Editor.

The reviewers have discussed the reviews with one another and the Reviewing Editor has drafted this decision to help you prepare a revised submission.

Summary:

The attenuation of probabilistic reward learning in older human participants was accompanied by reduced value signals in prefrontal cortex.

Essential revisions:

As you will see from the reviewers comments, which are backed by similar concerns of the Reviewing Editor, the paper is far too complicated as it stands. I would suggest to seriously reduce unnecessary analyses, and focus and streamline the paper, and its text, on the essentials related to the age of the participants. There is nothing wrong with removing parts of the data and/or analysis if that would lead to a much clearer message (note that the Abstract already is tough to read with too many distinct details). It should also be discussed why there were not the usual reward prediction error signals found in the ventral striatum (this is not necessarily a bad thing, in particular when a stringent analysis has been applied as here, but readers want to know why).

We will need to send the paper back to the reviewers, but given their substantial difficulties with reading and commenting on the complex text, we can do this only once. This one-revision-only is also general policy of the Journal, and we will need to adhere to it given the complexity of the report.

Please reply with a succinct, simple, point-to-point text to the reviewers' comments. And please be aware of a general policy at *eLife* that we do not permit several rounds of revisions. Thus, we sincerely hope that you will be able to successfully revise your manuscript with the next round.

*Reviewer #1:*

This study examined the behavioral and neural bases of age differences in probabilistic reward learning, using fMRI and PET. A group of young and older adults performed a simple instrumental reward learning task (two-armed bandit) in an fMRI experiment. On each trial, participants chose between two cues, whose reward probabilities changed in Gaussian random-walk processes. DA D1 binding potential (BP) was also assessed in several brain regions. Young adults made more money and more efficient choices. The authors compared two families of behavioral models, one based on reinforcement learning through reward prediction errors (RPEs), and the other on a Bayesian observer, where reward probability is updated after each outcome. The best model was a Bayesian one, which included reward-probability updating for both the chosen and unchosen options, the variance of the option not chosen on the previous trial, and decision confidence. None of the model parameters differed between the young and older adult groups, but using the model's generated expected values the authors show that young adults made more "adaptive switches" – switches to the option of the higher value. The winning model was used in the analysis of the fMRI data. This analysis revealed stronger representation of the expected value of the chosen option in young compared to older adults in several brain areas. In the vmpfc, this parameter predicted earnings, and accounted for age differences in aging. DA D1 BP in NAcc was correlated with the value parameter in vmpfc, but not in NAcc, and accounted for age differences in that parameter. The authors then searched for RPE signals. They did not settle for a simple correlation with RPE, but rather required separate correlations with expected and obtained reward, with opposite signs. This analysis did not identify RPE signals in NAcc in either young or older adults, but the authors show that a reinforcement-learning model fits the BOLD data there better than the Bayesian model. Finally, the authors also report activation in several areas that are related to decision confidence or to switches.

This is an interesting study, which asks an important question. There is evidence for reduced reinforcement learning in aging, but the neural basis of this reduction is not clear. The paper has many strengths, including the use of computational modeling and model comparison, the combination of fMRI and DA D1 BP within subject, and the careful neural analysis. I have relatively minor comments, which are detailed below.

– In its current form the paper is somewhat difficult to follow. There are many questions and analyses, so the writing should be very clear in order to take the reader through the entire story. Sometimes the authors assume knowledge that a wide audience may not necessarily have. For example, will be helpful if there is a brief description of the task either at the end of the Introduction or the beginning of the Results section. Then perhaps some overview of the main questions and the general analysis strategy. Next, the model descriptions should be clarified – the Materials and methods section provides detailed information, but it will be helpful if the brief description in the Results section is clearer – especially important is the definition of all the parameters in each model. Keeping all the parts of the PET analysis together will also be helpful. Finally, it seems that the main finding is the relationships between vmPFC activity and behavior and between NAcc DA BP and vmPFC activity. In both of these analysis correlation with age disappears when the physiological variable is taken into account. This is very interesting and should be clearly stated.

– It was a bit confusing to me that no parameter of the winning model reflected the age-related difference in behavior (unlike the RW model). It seems that the main point of using computational modeling is to uncover latent variables that affect behavior, but cannot be directly observed, in order to understand the differences in computations. Is it possible that the model fails to capture some latent variable that is of the most interest to this particular study? This is supported by the fact that using the switches, instead of the model parameters, yielded more informative results. The authors should clearly explain the utility of model fitting for their behavioral analysis. In particular, did the vmPFC results depend on the particular formulation of Q from the winning model?

– The lack of RPE encoding in NAcc in young adults is presented as an incidental finding, but it is of importance, independently from the aging research question. The authors are right that this may be due to their stringent criterion, but the fact that there was also no correlation with D1 BP, and that an RW model fit the activity better than the winning Bayesian model, makes me wonder if the Q estimates may be off? Also, is there an age difference if you consider the less stringent single-predictor RPE?

*Reviewer #2:*

In this article, the authors combine behavioral, fMRI, PET, and computational modeling approaches to understand the mechanisms of probabilistic reward learning, and how this learning changes with age. There are definitely some interesting results here. The relationship between D1 binding potential (BP) in NAcc and the neural correlate of chosen value in vmPFC seems particularly notable. However, several of the neural and computational modeling results are not yet as compelling as they could be. Below the major findings are discussed in turn; in some cases the paper might be best served by cutting certain analyses entirely, but suggestions and comments are provided nonetheless.

1) Probably the most novel and central findings concern predicted value (Q) signals in vmPFC. The correlation between Q and vmPFC activity is reduced in older adults and predicted by D1 BP in NAcc, and D1 BP fully accounts for the effect of age. The predicted value response in vmPFC also predicts performance on the task. This is an interesting set of findings. I'm aware of only one other report showing age effects on vmPFC value correlates (Halfmann et al., 2017, SCAN), but that report focused on individual differences within older adults, and the link to dopaminergic signal shown here provides a plausible mechanism for the effect. These findings could be strengthened in a few ways, however:

1a) A formal mediation analysis would further strengthen the claim that D1 BP accounts for the effects of age on value signals in vmPFC.

1b) These results depend on the parameter estimates in vmPFC extracted from the region showing a main effect of predicted value. It would be of interest to replicate these analyses in an independent ROI – e.g., the ROI from the Bartra et al., 2013 meta-analysis on subjective value. Though the age comparison is orthogonal to the original fMRI analysis, it is hard to know if and how the possible inflation of the parameter estimates might interact with other analyses such as the correlation with D1 BP.

1c) The effect of Q in vmPFC on performance in the task is significant when controlling for age and model fit, but does it hold when not controlling for these factors? I understand the need to control for age given differences in value-related vmPFC activity between the two groups, but the results without the control variables should at least be noted in the text.

1d) Given the high correlation between dopamine binding in different ROIs, theorizing of how D1 binding in the NAcc specifically could mediate vmPFC effects (Discussion section) seems somewhat premature.

1e) In these analyses, a negative correlation with predicted value is also noted in several prefrontal and parietal regions. In several previous studies, these regions have been associated with difficult choices, indexed by the absolute difference between chosen and unchosen value. Before concluding that these regions encode the inverse of chosen value, this alternative explanation would need to be ruled out.

2) Another imaging finding was that NAcc tracked received rewards, rather than reward prediction errors, in both young and old adults. This is an interesting finding and the methods here provide a nice warning about making strong conclusions about correlations with prediction error regressors, without examining responsivity to both components of the prediction error.

2a) One possibility, though, is that the lack of a prediction error signal is reflective poor learning – i.e., subjects' expectancies are not accurate. Have the authors looked at the correlation between the representation of expectancy in NAcc and performance on the task?

2b) It looks like two different versions of Figure 4 were uploaded. Which one is correct needs to be clarified.

3) A final neural finding – which is more exploratory – is increased activity in frontoparietal brain regions on switch trials, which predicts performance in the task.

3a) Here again, the possible alternative that these regions respond to more difficult decisions (indexed, for example, but the difference in absolute value, or by reaction time), rather than switches per se, needs to be explored.

3b) In addition, the authors also show that activity in these regions is negatively related to the number of switches made by the subject, which is in turn, negatively related to performance. Does dlPFC or IPL activity predict performance after including the number of switches in the model? Without showing this, it is quite possible that the number of switches modulates both dlPFC/IPL activity and performance (i.e., the relationship between the brain and performance is driven by a third variable).

3c) While the fact that switch-related neural activity independently predicts performance when controlling for Q in vmPFC suggests that including the switch-related activity improves the predictive power of the model, a formal model comparison is needed to support this conclusion. I would like to see a formal model comparison between the following models for predicting performance: 1) age and Q in vmPFC; 2) age and switch-related activity in switch ROIs; and 3) age, Q in vmPFC, and switch-related activity in switch ROIs.

4) The results are not definitive on whether there are age differences in probabilistic learning and if so what the cause of these differences is.

4a) Performance differences between older and younger adults are only significant with a one-tailed t-test. This is weak evidence at best for any age effects in the task.

4b) None of the parameters in the authors' winning model differed between younger and older adults. The authors suggest that "correlated changes in the parameters may explain the age difference". However, if there were correlated changes, there should still be significant differences in the parameters – in fact, wouldn't you expect to see more significant differences? I suppose the authors could use some kind of multivariate analysis to look for age differences in model parameters, but the overall picture seems more consistent with subtle, if any, age effects.

5) There were several aspects of the computational modeling approach that were potentially problematic.

5a) In the authors' winning Bayesian model, Q values are initialized at 0.5 and the forgetting process relaxes these values back to 0.5. In the reinforcement learning models, Q values are initialized at 0 and the forgetting process relaxes these values back to 0. A fair comparison between the two classes of models would eliminate this structural difference. Though unlikely, it is possible that this aspect, rather than the details of updating, accounts for the difference in model performance between RL and Bayesian models.

5b) In the authors' winning model, switching is more likely when there is less uncertainty about the unchosen value and when there is greater relative confidence about the previous choice. These effects are counter-intuitive, and more evidence is needed for them to be convincing.

In the case of switching when there is less uncertainty, this is the opposite of normative exploration, the notion of an "exploration bonus." Wilson et al., 2014, for example, found evidence for directed exploration. Why do the authors think they see the opposite here?

5c) That previous trial relative confidence predicts switching is also surprising. It is hard to see how this could be a good feature for learning to have under general conditions, which makes me wonder if it is a byproduct of some particular aspect of the current task. For example, this behavior could be adaptive if there is a negative correlation between the values of the two options or a tendency for values to reverse over time. If this result is more of a byproduct of the task than a general phenomenon, then I would worry about making too much of this finding.

5d) In both of these cases, it would be very informative if the authors could identify the features of task performance that these two aspects of the model explain. This would increase confidence in the empirical finding, beyond the simple model comparisons. It might also provide further insight and modeling ideas; perhaps once the nature of switching behavior in this task is better understood the authors will discover that it can be even better explained by adding different, less counter-intuitive, features to the model.

5e) The potential interaction between the uncertainty and confidence effects needs to be examined. It would make sense that uncertainty about the unchosen value and relative confidence are negatively correlated, given that the former is one of the inputs needed to calculate the latter. This would seem to complicate any interpretation of the weights on these parameters when both are in the model. At a minimum, the authors should report the goodness of fit statistics and parameter weights for a model where only the relative confidence term, and not the uncertainty term, is included in the model.

5f) The authors refer to the effect of relative confidence as a "grass is greener" effect, but I do not think this analogy captures the effect accurately at all. For example, I could imagine also referring to an effect in the exact opposite direction (more switching when confidence is lower) as a "grass is greener" effect, so obviously the analogy is doing no work, and perhaps obscuring rather than enlightening.

*Reviewer #3:*

In this work, De Boer and colleagues examined the effects of age and dopamine (D1 receptor availability measured using PET) on the neural mechanisms underlying probabilistic reward learning (explored using fMRI). They isolated two main processes contributing to choice performance: 1) learned estimates of option values, 2) switching behavioral strategy. The first process was notably expressed in vmPFC activity and declined with age, this decline being related to nucleus accumbens dopamine. The second process was underpinned by a frontoparietal activity and was independent of age and dopamine.

The question is not really novel. In particular the last author (Marc Guitart-Masip) contributed to a Nature Neuroscience paper that already established the dopamine-dependency of age-related decline in reward learning. However, this new study brings further insights that help to refine our understanding of this phenomenon. Besides, the study has several strengths: it gathers a large dataset (60 participants) including behavioral, PET and fMRI data and takes a sophisticated analytical approach using computational modeling. Overall, I think this paper would nicely contribute to unraveling the determinants of reward learning in humans. Unfortunately, the number of different analyses and results sort of obscure the reading and dilute the main findings. My main suggestion would be to streamline the analysis so the results description would have a clearer structure.

In that regard, it would help to remove the Bayesian model, which does not seem to bring much to the main conclusions, unless I missed something. I appreciate the amount of effort that the authors must have invested in this modeling work, but I am not convinced it makes sense to keep this model and related analyses of brain activity. My reasons are 1) there is no principle justifying that participants should switch when confidence in the chosen option is high (I suspect this comes from correlation between parameters), 2) when comparison is fair (models without the confidence add-ons) the BIC of Rescorla-Wagner and Bayesian models are similar (compare third lines in Table 1), 3) unlike the RW model, the Bayesian model does not capture the difference in behavioral performance between young and older people, 4) the variables specific to the Bayesian model have only weak links with brain activity, contrary to the RW model-based predictions, on which main conclusions are built.

Besides, I have some other concerns:

– As far as I understand, a unique random walk was used to generate reward probabilities for all participants. From the plot in Figure 1 it looks like a noisy reversal, which raises the issue of possible age-related deficits in reversal per se, and of the anti-correlation between cues that may induce the belief that outcomes inform on both cues (subject might normalize the two option values). These possibilities should be discussed.

– The difference in vmPFC value signal could artificially come from the difference in learning performance. This is because the variance of the value regressor in the GLM used to fit fMRI data depends on how much subjects learn about option values (no learning gives a flat regressor), unless regressors are z-scored (I could not find this info in the Materials and methods). This issue needs to be carefully addressed.

– The absence of (negative) correlation with expectation at outcome onset is interesting given the debate about prediction error encoding in the striatum. Yet I am unsure of how the authors interpret this. Is this an artifact from the design (cue and outcome onsets being too close in time), is it that true prediction errors are encoded in other brain regions, or is it that the brain does not encode prediction error at all? Perhaps the authors could clarify their position in this issue in the discussion.

---

## [Author Response]

Reviewer #1:

*[…] I have relatively minor comments, which are detailed below.*
*– In its current form the paper is somewhat difficult to follow. There are many questions and analyses, so the writing should be very clear in order to take the reader through the entire story. Sometimes the authors assume knowledge that a wide audience may not necessarily have.*

We apologise for the confusion caused by the amount of information included in the paper. We thank the reviewer for providing very helpful structuring comments. We have done our best to streamline the paper by cutting out some of the fMRI analyses, and address the points made by the reviewer, one by one, below:

*For example, will be helpful if there is a brief description of the task either at the end of the Introduction or the beginning of the Results section.*

We have added a paragraph in the Introduction that explains the TAB in more detail:

“In brief, all participants performed 220 trials on the TAB (Figure 1). […] Participants received monetary earnings of 1 Swedish Krona (SEK, ~$0.11) per rewarded trial.”

Then perhaps some overview of the main questions and the general analysis strategy.

We have added a paragraph at the beginning of the Results section outlining the goal of the analyses and main questions:

“The goal of the analyses was to establish the neural mechanism underlying decreased probabilistic value learning in older participants. […] To obtain the best estimate of expected value to use in our fMRI analysis, we fitted a range of computational models and used Bayesian model selection.”

*Next, the model descriptions should be clarified – the Materials and methods section provides detailed information, but it will be helpful if the brief description in the Results section is clearer – especially important is the definition of all the parameters in each model.*

We have clarified this section in the Results, which currently reads as follows:

“The first determinant of switching was the current variance (V) of the option that was not chosen on the previous trial calculated from its approximate β distribution (Figure 2; formula 8, Materials and methods). […]Hence, increased uncertainty about the previously unchosen option caused most subjects to stick to their current choice.”

And:

“The second determinant of switching was a measure of the relative confidence in the choice that was made on the previous trial (see Materials and methods). […] Thus, if κwas positive, then a subject would be more likely to switch on trial if she had been more confident on trial t-1.”

We have also added a figure to clarify the components of the model (new Figure 2).

*Keeping all the parts of the PET analysis together will also be helpful.*

We have restructured the Results section to first present the behavioural and computational modelling results, then the analysis of vmPFC activity, then RPEs in the NAcc, and last the PET results. The results concerning neural correlates of confidence and switch behaviour are no longer part of the manuscript.

*Finally, it seems that the main finding is the relationships between vmPFC activity and behavior and between NAcc DA BP and vmPFC activity. In both of these analysis correlation with age disappears when the physiological variable is taken into account. This is very interesting and should be clearly stated.*

We have emphasised this in the Results by presenting the result as a mediation analysis. We added:

“The parameter estimate for Q in vmPFC was positively related to total monetary gains (r(53)=0.37, p=0.006, controlling for age and model fit in a partial correlation). […] This is consistent with a full mediation of age effects on performance by Q in vmPFC. Note however, that it is difficult to make inferences on mediation effects of age in a cross-sectional dataset (1).”

And:

“This result was confirmed by a mediation analysis: […] This is consistent with a full mediation of age effects on Q in vmPFC by DA D1 BP in NAcc.”

Finally, we point this result out again in the Discussion:

“Our results are consistent with a full mediation of the age effects on performance by Q in vmPFC, that is, age no longer predicts performance when controlling for the strength of BOLD that reflects Q in vmPFC. The same is true for the strength of Q in vmPFC: the effect of age can be explained by lower DA D1 BP in the older age group. Note however, that it is difficult to make inferences on mediation effects of age in a cross-sectional dataset (1).”

*– It was a bit confusing to me that no parameter of the winning model reflected the age-related difference in behavior (unlike the RW model). It seems that the main point of using computational modeling is to uncover latent variables that affect behavior, but cannot be directly observed, in order to understand the differences in computations. Is it possible that the model fails to capture some latent variable that is of the most interest to this particular study? This is supported by the fact that using the switches, instead of the model parameters, yielded more informative results. The authors should clearly explain the utility of model fitting for their behavioral analysis. In particular, did the vmPFC results depend on the particular formulation of Q from the winning model?*

We agree that the lack of difference in model parameters is unsettling. However, we believe that the Bayesian model provides the best account of our data. We set out to do computational modelling with three aims in mind. First, we aimed to uncover behavioural mechanisms on this particular task, regardless of age. We were successful in this regard by demonstrating that this model provides a better account of choices as indicated by Bayesian model comparison. Further, the model uncovers previously unclear contributions of uncertainty and confidence to choice on this task. For example, the model uncovers that perseveration is better accounted by as uncertainty aversion than by a choice kernel as previously thought. Our second goal was to generate predictors of neural responses. In this regard, we also observed that the Bayesian model provides a better predictor of expected value in the vmPFC improving our ability to make inferences about the relationship between BOLD signal in this region and dopamine D1 receptor availability (see below). We showed this by building two equivalent GLMs for fMRI analysis: they both include reward at the time of outcome, and expected value (as calculated by each respective model) at the time of choice. A paired t-test between the residuals of both GLMs demonstrates that the estimates for expected value from the Bayesian model predict BOLD in vmPFC more accurately than expected value estimates from the RW model. Our third goal was to understand group differences in behaviour. In this respect, we were not successful in the sense that the Bayesian model failed to capture the group difference we observed in behaviour (that was captured by the RW model). We state this limitation in the Discussion (Also see reviewer 2, point 4b and 6k and reviewer 3, point 3):

In fact, it is likely that the process underlying age differences in performance is not parametrised in the winning Bayesian model.

We add the observation that the Bayesian model generates better predictions of BOLD in vmPFC at choice in the Results section along with the previously reported observation that the RW model generates better predictions of BOLD in NAcc at outcome (Also see reviewer 2 point 4b and reviewer 3 point 3):

“On the other hand, the Bayesian observer model generates better predictions of the BOLD signal in the vmPFC when Q as generated by each model was included as a parametric modulator at the time of choice (paired t-test comparing residuals of the respective GLM models across all voxels in the respective vmPFC ROIs; t(56)=5.62, p<0.001).”

Our vmPFC result is not dependent on the choice of model. When expected value is estimated using the RW model, the group difference in anticipatory expected value in the vmPFC and its relationship to performance remains unchanged. We have added this result:

“The results were not dependent on the use of the Bayesian model to estimate Q values (when using the RW model Q estimates; when including both age and Q, β age = -0.20, -0.22, -0.21, p=0.111, 0.093, 0.104; β Q in vmPFC = 0.33, 0.28, 0.26, p=0.010, p=0.030, p=0.047 for monetary gains, efficient choices and adaptive switches, respectively).”

The relationship between this activity and DA D1 BP is still significant, but does not survive correction for age. Interestingly, the relationship between age and anticipatory activity in vmPFC does not survive the inclusion of DA D1 either. This does not allow for the disentanglement of age and DA D1 BP as contributors to vmPFC activity when using the RW model. However, since the BOLD activity in vmPFC is better accounted for by the Bayesian model, which is also a better model in terms of Bayesian model comparison, we see this as a good reason to keep the Bayesian model as a predictor of brain activity. Note that despite the RW generating better predictions for BOLD signal in the NAcc, the reported lack of RPE is not dependent on which model is used.

*– The lack of RPE encoding in NAcc in young adults is presented as an incidental finding, but it is of importance, independently from the aging research question. The authors are right that this may be due to their stringent criterion, but the fact that there was also no correlation with D1 BP, and that an RW model fit the activity better than the winning Bayesian model, makes me wonder if the Q estimates may be off? Also, is there an age difference if you consider the less stringent single-predictor RPE?*

There is no significant difference between the signal for Q in NAcc from the GLM where Q is estimated from the Bayesian model compared to the Rescorla-Wagner model (paired t-test, p>0.165 for both sides). In addition, there is no difference between the age groups when the standard more liberal analysis (using a single RPE regressor) is used and regardless of whether expected value is estimated using the RW model or the Bayesian model (p>0.45 in all comparisons). We have added this negative result in the Results section:

“In addition, when performing a less stringent test and extracting parameter estimates from this ROI for the full RPE, defined as one regressor (R-Q), we did not observe any differences between the groups’ mean activation (p>0.45).”

Reviewer #2:

*In this article, the authors combine behavioral, fMRI, PET, and computational modeling approaches to understand the mechanisms of probabilistic reward learning, and how this learning changes with age. There are definitely some interesting results here. The relationship between D1 binding potential (BP) in NAcc and the neural correlate of chosen value in vmPFC seems particularly notable. However, several of the neural and computational modeling results are not yet as compelling as they could be. Below the major findings are discussed in turn; in some cases the paper might be best served by cutting certain analyses entirely, but suggestions and comments are provided nonetheless.*

We thank reviewer 2 for their insightful comments and for the very detailed comments on our manuscript. We apologise for any lack of clarity in the original version, and believe the quality of our manuscript has greatly improved thanks to these comments.

*1) Probably the most novel and central findings concern predicted value (Q) signals in vmPFC. The correlation between Q and vmPFC activity is reduced in older adults and predicted by D1 BP in NAcc, and D1 BP fully accounts for the effect of age. The predicted value response in vmPFC also predicts performance on the task. This is an interesting set of findings. I'm aware of only one other report showing age effects on vmPFC value correlates (Halfmann et al., 2017, SCAN), but that report focused on individual differences within older adults, and the link to dopaminergic signal shown here provides a plausible mechanism for the effect. These findings could be strengthened in a few ways, however:*

*1a) A formal mediation analysis would further strengthen the claim that D1 BP accounts for the effects of age on value signals in vmPFC.*

We have now presented this result as a formal mediation analysis (Also see reviewer 1, first concern, last bullet point).

“This result was confirmed by a mediation analysis: Age was a significant predictor of both BP in NAcc (r(54)=-0.78, p<0.001) and Q in vmPFC (r(55)=-0.32, p=0.016). BP in Nacc was also a significant predictor of Q in vmPFC r(54)=0.41, p=0.001. Age was no longer a significant predictor of Q in vmPFC after controlling for BP in NAcc (β age=-0.01, p=0.964; β BP in NAcc=0.42, p=0.038).”

*1b) These results depend on the parameter estimates in vmPFC extracted from the region showing a main effect of predicted value. It would be of interest to replicate these analyses in an independent ROI – e.g., the ROI from the Bartra et al., 2013 meta-analysis on subjective value. Though the age comparison is orthogonal to the original fMRI analysis, it is hard to know if and how the possible inflation of the parameter estimates might interact with other analyses such as the correlation with D1 BP.*

We acknowledge that it is possible that these values are inflated. Therefore, we performed the suggested analysis, on the ROI resulting from their five-way conjunction analysis (Figure 9 in said paper), carrying a monotonic, modality-independent subjective value signal. This analysis demonstrated that the relationship between Q in the vmPFC and DA D1 BP indeed survived when using the suggested ROI: partial correlation between activity in the Bartra 2013 ROI and DA D1 BP: r(53)=0.318, p=0.018 (controlled for age and model fit, also survives without controlling for these variables). The correlation between Q in vmPFC and performance also survived when using the suggested ROI: partial correlation between activity in the Bartra 2013 ROI and monetary gains: r=0.337, p=0.011 (controlling for age and model fit, also survives without controlling for these).

*1c) The effect of Q in vmPFC on performance in the task is significant when controlling for age and model fit, but does it hold when not controlling for these factors? I understand the need to control for age given differences in value-related vmPFC activity between the two groups, but the results without the control variables should at least be noted in the text.*

The relationship between these variables hold when not controlling for these factors, r=0.46, p<0.001. We have added this in the text:

“The parameter estimate for Q in vmPFC was positively related to total monetary gains (r(53)=0.37, p=0.006, controlling for age and model fit in a partial correlation). This correlation remained significant without controlling for age, model fit or both.”

*1d) Given the high correlation between dopamine binding in different ROIs, theorizing of how D1 binding in the NAcc specifically could mediate vmPFC effects (Discussion section) seems somewhat premature.*

We thank the reviewer for pointing this out. It is correct that BP in the different ROIs are correlated and we added a note on that in the Discussion.

However, BP in NAcc is the only measure for which a mediation analysis is significant. This is consistent with the literature on reward processing in the corticostriatal loops. We therefore think this is important to discuss.

“Although BPs are highly correlated across ROIs, a mediation analysis was only significant for the NAcc. This is compatible with the literature on reward processing in the corticostriatal loops.”

*1e) In these analyses, a negative correlation with predicted value is also noted in several prefrontal and parietal regions. In several previous studies, these regions have been associated with difficult choices, indexed by the absolute difference between chosen and unchosen value. Before concluding that these regions encode the inverse of chosen value, this alternative explanation would need to be ruled out.*

This is indeed a very valid point. For the sake of clarity and brevity, we have taken out these analyses.

*2) Another imaging finding was that NAcc tracked received rewards, rather than reward prediction errors, in both young and old adults. This is an interesting finding and the methods here provide a nice warning about making strong conclusions about correlations with prediction error regressors, without examining responsivity to both components of the prediction error.*
*2a) One possibility, though, is that the lack of a prediction error signal is reflective poor learning – i.e., subjects' expectancies are not accurate. Have the authors looked at the correlation between the representation of expectancy in NAcc and performance on the task?*

This is a good point. The correlation between performance and Q in NAcc is not significant (p>0.25 in all correlations, with or without controlling for age). We have added this negative result in the manuscript:

“There was no indication that the lack of expected value signal in the NAcc at the group level was caused by some participants showing poor learning of expected value, as the correlation between Q in NAcc and the different measures of performance (monetary gains, effective choices, and adaptive switches) was not significant (p>0.25).”

*2b) It looks like two different versions of Figure 4 were uploaded. Which one is correct needs to be clarified.*

These are the RW and the Bayesian RPE parameter estimates – one of them is a supporting figure. This is now clarified in the captions and text references.

“Extracted parameter estimates for R and Q as calculated by the Bayesian observer model from the regions shown in Figure 4. Although we found a strong effect of reward bilaterally, no expected-value signal was observed for either age group (p>0.10).”

*3) A final neural finding – which is more exploratory – is increased activity in frontoparietal brain regions on switch trials, which predicts performance in the task.*

We would like to thank the reviewer for the insightful comments on this section, which has led us to cut out the bulk of these analyses.

*3a) Here again, the possible alternative that these regions respond to more difficult decisions (indexed, for example, but the difference in absolute value, or by reaction time), rather than switches per se, needs to be explored.*

We thank the reviewer for this important point. We have performed the suggested analysis including both RT and value difference) and although significant clusters are still obtained in IPL and OFC, the relationship between activity in these clusters and performance has disappeared. Therefore, and for the sake of improving and streamlining the paper, we have taken out this analysis.

*3b) In addition, the authors also show that activity in these regions is negatively related to the number of switches made by the subject, which is in turn, negatively related to performance. Does dlPFC or IPL activity predict performance after including the number of switches in the model? Without showing this, it is quite possible that the number of switches modulates both dlPFC/IPL activity and performance (i.e., the relationship between the brain and performance is driven by a third variable).*

Yes, even when controlling for switches this activity predicts performance. However, because the relationship does not hold when including measures of difficulty in the GLM, and for the sake of streamlining the paper, we have now taken out the switch analyses.

*3c) While the fact that switch-related neural activity independently predicts performance when controlling for Q in vmPFC suggests that including the switch-related activity improves the predictive power of the model, a formal model comparison is needed to support this conclusion. I would like to see a formal model comparison between the following models for predicting performance: 1) age and Q in vmPFC; 2) age and switch-related activity in switch ROIs; and 3) age, Q in vmPFC, and switch-related activity in switch ROIs.*

The switch analysis on BOLD has now been removed, and only the Q in vmPFC model is presented in the paper.

*4) The results are not definitive on whether there are age differences in probabilistic learning and if so what the cause of these differences is.*

We respectfully disagree with the reviewer and outline our reasons for this below.

*4a) Performance differences between older and younger adults are only significant with a one-tailed t-test. This is weak evidence at best for any age effects in the task.*

We apologise for the lack of clarity on the performance difference. Apart from the weak age group difference on total wins, we also find difference between efficient choices and adaptive switches (reported in the Results and Figure 1). We had chosen total monetary gains as an indicator of performance because it is the most intuitive indicator of performance. However, the mediation of the effect of age on performance holds when other measures of performance that we present in the paper (efficient choices, adaptive switches) are used as the outcome variable. We have added a paragraph about this in the manuscript:

“Q in vmPFC was a significant predictor of all measures of performance (bivariate correlations: total monetary gains: r(55)=0.47, p<0.001 adaptive switches: r(55)=0.39, p=0.003; efficient choices: r(55)=0.38, p=0.004). […] Note however, that it is difficult to make inferences on mediation effects of age in a cross-sectional dataset (1).”

*4b) None of the parameters in the authors' winning model differed between younger and older adults. The authors suggest that "correlated changes in the parameters may explain the age difference". However, if there were correlated changes, there should still be significant differences in the parameters – in fact, wouldn't you expect to see more significant differences? I suppose the authors could use some kind of multivariate analysis to look for age differences in model parameters, but the overall picture seems more consistent with subtle, if any, age effects.*

We agree that the lack of difference in model parameters is unsettling. As the reviewer suggested, we performed a multivariate analysis with the parameters of the behavioural model as independent variables, and age group as a fixed factor. This analysis did not detect any effect of age group (F=0.91, p=0.482). This negative result is now reported:

“A multivariate analysis with the model parameters as independent variables, and age group as a fixed factor, did not yield any significant predictor of age group (F=0.91, p=0.482).”

However, we believe that the Bayesian model provides the best account of our data. We set out to do computational modelling with three aims in mind. First, we aimed to uncover behavioural mechanisms on this particular task, regardless of age. We were successful in this regard by demonstrating that this model provides a better account of choices, as indicated by Bayesian model comparison. Further, the model uncovers previously unclear contributions of uncertainty and confidence to choice on this task. For example, the model uncovers that perseveration is better accounted by as uncertainty aversion than by a choice kernel as previously thought. Our second goal was to generate predictors of neural responses. In this regard, we also observed that the Bayesian model provides a better predictor of expected value in the vmPFC improving our ability to make inferences about the relationship between BOLD signal in this region and dopamine D1 receptor availability (see below). We showed this by building two equivalent GLMs for fMRI analysis: they both include reward at the time of outcome, and expected value (as calculated by each respective model) at the time of choice. A paired t-test between the residuals of both GLMs demonstrates that the estimates for expected value from the Bayesian model predict BOLD in vmPFC more accurately than expected value estimates from the RW model. Our third goal was to understand age group differences in behaviour. In this respect, we were not successful in the sense that the Bayesian model failed to capture the age group difference we observed in behaviour (that was captured by the RW model).

We add the observation that the Bayesian model generates better predictions of BOLD in vmPFC at choice in the Results section along with the previously reported observation that the RW model generates better predictions of BOLD in NAcc at outcome (Also see reviewer 1, point 2, and reviewer 3, point 3):

“On the other hand, the Bayesian observer model generates better predictions of the BOLD signal in the vmPFC when Q as generated by each model was included as a parametric modulator at the time of choice (paired t-test comparing residuals of the respective GLM models across all voxels in the respective vmPFC ROIs; t(56)=5.62, p<0.001).”

*5) There were several aspects of the computational modeling approach that were potentially problematic.*
*5a) In the authors' winning Bayesian model, Q values are initialized at 0.5 and the forgetting process relaxes these values back to 0.5. In the reinforcement learning models, Q values are initialized at 0 and the forgetting process relaxes these values back to 0. A fair comparison between the two classes of models would eliminate this structural difference. Though unlikely, it is possible that this aspect, rather than the details of updating, accounts for the difference in model performance between RL and Bayesian models.*

This is a good point. We have refitted the RW model with starting values 0.5 and allowing the values to relax back to 0.5 as well. This model is better than the originally reported RW model (BIC: 10355 vs 10388) and has now been added to the manuscript in substitution of the previously reported RW (Table 1). The Materials and methods section has been updated to reflect this. However, this new RW model shows no better fit than the Bayesian model including the effect of variance of the unchosen option (BIC: 10335) or the winning full Bayesian model including both the effect of variance of the unchosen option and relative confidence (BIC: 10259) (Table 1).

*5b) In the authors' winning model, switching is more likely when there is less uncertainty about the unchosen value and when there is greater relative confidence about the previous choice. These effects are counter-intuitive, and more evidence is needed for them to be convincing.*
*In the case of switching when there is less uncertainty, this is the opposite of normative exploration, the notion of an "exploration bonus." Wilson et al., 2014, for example, found evidence for directed exploration. Why do the authors think they see the opposite here?*

We do not think that the effect of uncertainty that we observed is that counterintuitive. The reviewer is right that the effect of uncertainty we observed is opposite to an exploration bonus or uncertainty based exploration as suggested by normative accounts of exploration (2) and supported by some experiments (3,4). However, in many other studies of decision-making, variance is penalised as a form of risk sensitivity (5-7) which is akin to the effect that we observed. Furthermore, our model comparison showed that uncertainty aversion is a better account of the perseveration typically observed in bandit tasks (8,9) than a choice kernel. We have added this rationale in the Discussion:

“This is opposite to an exploration bonus or uncertainty based exploration term that arises in various more or less normative accounts of exploration (2) and has been observed in some experiments (3,4). However, many previous studies of decision-making have also shown that variance may be penalised as a form of risk sensitivity (5–7), and this is a cousin of the effect that we observed. Furthermore, our model comparison showed that uncertainty aversion is a better account of the perseveration typically observed in bandit tasks (8,9) than a choice kernel.”

*5c) That previous trial relative confidence predicts switching is also surprising. It is hard to see how this could be a good feature for learning to have under general conditions, which makes me wonder if it is a byproduct of some particular aspect of the current task. For example, this behavior could be adaptive if there is a negative correlation between the values of the two options or a tendency for values to reverse over time. If this result is more of a byproduct of the task than a general phenomenon, then I would worry about making too much of this finding.*

The effects of relative confidence observed at the group level are indeed surprising (certainly, we had not predicted them), and could partly stem from a particular aspect of the task that we have yet to identify. However, when looking at individual differences we observed a negative correlation between κ and total monetary gains on the task (-0.42, p<0.001). Note that this correlation was wrongly described in the Results section before (our apologies for this, see response to point 6b for details). Further, those participants that had a negative κ (8 out of 57) performed best on the task. This implies that relative confidence has the expected effect on performance despite having an unexpected sign at the group level. This effect of κ on performance can be observed from simulated data where we explored the effects of varying κ when all other parameters were fixed at the median of all participants. We plotted the mean and standard error for total wins and proportion efficient choices as a result of 100 iterations of 220 trials in the graph below

We are also curious about what causes this unexpected use of confidence in most participants and are planning experiments where we will manipulate different task features such as the volatility of the environment and beliefs about task structure to tease apart the mechanism by which confidence operates. We have added clarifying text in the Results section:

“Nevertheless, κ was negatively correlated with the total monetary gains on the task (r(54)=0.42, p=0.001, controlled for age; Supplementary Table 1), with negative values of κ in those participants with the highest performance. This implies that κ has the expected effect on performance despite having an unexpected sign at the group level.”

*5d) In both of these cases, it would be very informative if the authors could identify the features of task performance that these two aspects of the model explain. This would increase confidence in the empirical finding, beyond the simple model comparisons. It might also provide further insight and modeling ideas; perhaps once the nature of switching behavior in this task is better understood the authors will discover that it can be even better explained by adding different, less counter-intuitive, features to the model.*

From the model comparison (Table 1) it is evident that V (the variance of the unchosen option) captures perseveration better than RW or Bayesian models with a choice kernel. Therefore, the behavioural performance feature that is explained by V is perseveration. We provide a mechanistic account of perseveration beyond the commonly used choice kernel and show that, in our task, perseveration is steered by the variance of the previously unchosen option. We have highlighted this point in the current version of the task:

“This is a novel insight into the mechanism behind what is usually referred to as perseveration and suggests that aversion to the uncertainty about the option that was not chosen previously causes a tendency to stick to one choices.”

The performance feature captured by C^rel^ is less clear, and difficult to assess with the current dataset, which was not designed to find, let alone unpack, this quantity. Nevertheless, we have some ideas about it. One possibility is that κ depends on the perceived volatility of the task: if participants perceive the environment as being very volatile, an increased confidence in the currently chosen option can spur the belief that switching is a good idea. This could be assessed by manipulating volatility and/or regularly measuring experienced volatility, and observing the effect of κ on choice. Alternatively, it could be a belief in the Machiavellian nature of the experimenter (the surer a participant is that one option is better, the more likely they believe it will switch). Finally, the observed effect of κ could reflect a sort of safe exploration – in two senses: a) – the participant is convinced she has recently chosen the best option a lot (hence her confidence), so she can afford the odd exploratory trial; b) since the participant is relatively sure about the quality of A, they don't have to maintain a very precise assessment of by how much it bests B, so choosing B isn't informationally tricky for them in the sense of allowing a single outcome for B incorrectly to sway choice in favour of this option. In sum, we believe that this result is counterintuitive but very suggestive and points to new avenues for research. We have developed the discussion of this finding:

“One reason for the unwarranted use of confidence in the majority of participants could be that participants perceived the task as being highly volatile. As a result, they may have inferred that increasing confidence in the most recent choice indicates that the unchosen option has become better than the chosen option (10,11). Additionally, the observed effect of κ could reflect safe exploration: if the participant is convinced they have recently chosen the better option frequently (hence their confidence), they can afford to explore the more uncertain option. These possibilities provide interesting directions for future research.”

*5e) The potential interaction between the uncertainty and confidence effects needs to be examined. It would make sense that uncertainty about the unchosen value and relative confidence are negatively correlated, given that the former is one of the inputs needed to calculate the latter. This would seem to complicate any interpretation of the weights on these parameters when both are in the model. At a minimum, the authors should report the goodness of fit statistics and parameter weights for a model where only the relative confidence term, and not the uncertainty term, is included in the model.*

It is true that the correlation between υ(unchosen) and κ is negative, but this correlation is only significant at trend level (r(57)=-0.253, p=0.057). The models with only κ or only υ(unchosen) show poorer fit than the model with both υ(unchosen) and κ. We have added the model statistics for the model with κ only into Table 1 (likelihood: -5675.3, pseudo-R^2^: 0.331, iBIC: 11426). When the models are specified with one of the parameters alone, the sign of these parameters are largely the same as they are in the model with both parameters. In other words, if a Bayesian model is specified including κ only, κ is positive for 42 out of 57 participants (compared to 49 out of 57 in the current winning model), where all 42 participants with positive κ in the simpler model have positive κ in the winning model as well. If a Bayesian model is specified including υ only, υ is negative for 53 out of 57 participants (compared to 55 out of 57 in the current winning model), where all 53 participants with negative υ in the simpler model have negative υ in the winning model as well. This suggests that overall tendency for κ to be positive and υ to be negative does not stem from autocorrelation between the two. We have added this information in the Results (Also see reviewer 3, point 1):

“The overall tendency for κ to be positive and υ to be negative does not stem from autocorrelation between the two as the sign of these parameter is largely the same when the model is specified with only one of these parameters (data not shown).”

*5f) The authors refer to the effect of relative confidence as a "grass is greener" effect, but I do not think this analogy captures the effect accurately at all. For example, I could imagine also referring to an effect in the exact opposite direction (more switching when confidence is lower) as a "grass is greener" effect, so obviously the analogy is doing no work, and perhaps obscuring rather than enlightening.*

We apologise for the confusion created and have removed the expression to avoid ambiguity.

Reviewer #3:

*[…] The question is not really novel. In particular the last author (Marc Guitart-Masip) contributed to a Nature Neuroscience paper that already established the dopamine-dependency of age-related decline in reward learning. However, this new study brings further insights that help to refine our understanding of this phenomenon. Besides, the study has several strengths: it gathers a large dataset (60 participants) including behavioral, PET and fMRI data and takes a sophisticated analytical approach using computational modeling. Overall, I think this paper would nicely contribute to unraveling the determinants of reward learning in humans. Unfortunately, the number of different analyses and results sort of obscure the reading and dilute the main findings. My main suggestion would be to streamline the analysis so the results description would have a clearer structure.*

We thank the reviewer for highlighting the strength of the results and the constructive criticism, which has been very helpful in the process of streamlining our analyses. We apologise for our lack of clarity. Below we address each of the separate points.

*In that regard, it would help to remove the Bayesian model, which does not seem to bring much to the main conclusions, unless I missed something. I appreciate the amount of effort that the authors must have invested in this modeling work, but I am not convinced it makes sense to keep this model and related analyses of brain activity.*

We have streamed the result as suggested by the reviewer but, with due respect, decided to keep the Bayesian model. The reasons for this decision are outlined in response to each of the reviewer’s points pertaining this concern.

*My reasons are 1) There is no principle justifying that participants should switch when confidence in the chosen option is high (I suspect this comes from correlation between parameters),*

When the models are specified with one of the parameters alone, the sign of these parameters are largely the same as they are in the model with both parameters. In other words, if a Bayesian model is specified including κ only, κ is positive for 42 out of 57 participants (compared to 49 out of 57 in the current winning model), where all 42 participants with positive κ in the simpler model have positive κ in the winning model as well. If a Bayesian model is specified including υ only, υ is negative for 53 out of 57 participants (compared to 55 out of 57 in the current winning model), where all 53 participants with negative υ in the simpler model have negative υ in the winning model as well. This suggests that overall tendency for κ to be positive and υ to be negative does not stem from autocorrelation between the two. We have added this information in the Results (Also see reviewer 2 point 5e):

“The overall tendency for κ to be positive and υ to be negative does not stem from autocorrelation between the two as the sign of these parameter is largely the same when the model is specified with only one of these parameters (data not shown).”

2) When comparison is fair (models without the confidence add-ons) the BIC of Rescorla-Wagner and Bayesian models are similar (compare third lines in Table 1),

This is true and indicates that the reason why the Bayesian model is better is not dependent on the update rules. Instead, the Bayesian model provides additional information such as variance and confidence that can be used to make decisions. The Bayesian model has access to this information by tracking the probability distribution of the bandits; the simpler, RW, model does not (at least not simply). What makes the Bayesian model better, in our view, is that it provides the most parsimonious account of the behaviour we observed. In addition, it provides a mechanistic account of perseveration, commonly observed in bandit tasks. It also provides insight into the role of confidence, which is novel and can be interesting for future research. Finally, the BOLD activity in vmPFC is better accounted for by the Bayesian model (see response to point 4 for further discussion on this issue).

*3) Unlike the RW model, the Bayesian model does not capture the difference in behavioral performance between young and older people,*

This is true and unfortunate. Although one goal of using modelling is to better understand age group differences in behaviour, another goal with the modelling to uncover behavioural mechanisms on this particular task, regardless of age. We believe that the Bayesian model provides a better account of choices as indicated by Bayesian model comparison. Furthermore, the Bayesian model uncovers potentially interesting contributions of uncertainty and confidence to choice on this task such as that perseveration is better accounted of by uncertainty aversion than by a choice kernel as commonly modelled. We have added in the Discussion a sentence acknowledging that the components of our model may not be able to capture age differences (Also see reviewer 2, point 6k):

“In fact, it is likely that the process underlying age differences in performance is not parametrised in the winning Bayesian model.”

*4) The variables specific to the Bayesian model have only weak links with brain activity, contrary to the RW model-based predictions, on which main conclusions are built.*

The reviewer is right that the variables specific to the Bayesian model have only weak links with brain activity. We have therefore taken out the analyses that show the relation of BOLD to confidence and variance. We have reanalysed our fMRI data using a GLM only including Q at choice (as opposed to Q, V and Crel). Our most interesting results relate to expected value and do not change using this new GLM. We now use the parameter estimates of this new model for expected value to analyse the relationship between Q, age and DA. We observed that the Bayesian model provides a better predictor of expected value in the vmPFC compared to the RW model, improving our ability to make inferences about the relationship between BOLD signal in this region and dopamine D1 receptor availability. This was demonstrated by a paired t-test between the residuals of both GLMs in their respective vmPFC ROIs. We have included this observation in the Results (Also see reviewer 1, second concern, and reviewer 2, point 4b).

“On the other hand, the Bayesian observed model generates better predictions of the BOLD signal in the vmPFC when Q as generated by each model was included as a parametric modulator at the time of choice (paired t-test comparing residuals of the respective GLM models across all voxels in the respective vmPFC ROI; t(56)=-5.62, p<0.001).”

Whereas the RW model provides a better predictor of expected value in the NAcc, the reported lack of RPE is not dependent on which model is used (Figure 4—figure supplement 1).

It is important to note that our vmPFC result is not dependent on the choice of model. When expected value is estimated using the RW model, the age group difference in anticipatory expected value in the vmPFC and its relationship to performance remains unchanged. The relationship between this activity and DA D1 BP is still significant, but does not survive correction for age. Nevertheless, the relationship between age and anticipatory activity in vmPFC does not survive the inclusion of DA D1 either. This does not allow for the disentanglement of age and DA D1 BP as contributors to vmPFC activity when using the RW model. By contrast, BOLD activity in vmPFC is better accounted for by the Bayesian model, which in turn is also a better model (taking appropriate account of the numbers of parameters). We suggest that these are good reasons to keep the Bayesian model as a predictor of brain activity.

*Besides, I have some other concerns:*
*– As far as I understand, a unique random walk was used to generate reward probabilities for all participants. From the plot in Figure 1 it looks like a noisy reversal, which raises the issue of possible age-related deficits in reversal per se, and of the anti-correlation between cues that may induce the belief that outcomes inform on both cues (subject might normalize the two option values). These possibilities should be discussed.*

The reviewer is right that the TAB task can be seen as a noisy reversal learning task whereby the key determinant of which stimulus is chosen is expected value. However, as shown by our winning model, choice is also modulated by uncertainty and relative confidence.

Our results demonstrate that performance in the task is at least partly supported by the expected value signal in the vmPFC and that the strength of this signal explains the effects of age on performance. However, as pointed out by the reviewer, other mechanisms such as executive control may be at play. We attempted to uncover one such alternative mechanism by looking at brain responses on switch trials. However, after controlling for RT and value difference (as suggested by reviewer 2) these brain responses were no longer correlated with measures of performance, suggesting that they may be related to the differences in value. We have removed these results from the manuscript but extended the discussion to suggest that differences in executive functions such as the ability to inhibit a response to previously rewarded option could also be at play in our task because of the fact that the task can be seen as a noisy reversal learning task:

“Our results show that performance in the TAB is supported by the expected value signal in the vmPFC and that the strength of this signal explains the effects of age on performance. However, considering that the TAB can be seen as noisy reversal learning task, it is a possibility that differences in executive functions – such as the ability to inhibit a response to previously rewarded option – contribute to age group differences in our task (19).”

*– The difference in vmPFC value signal could artificially come from the difference in learning performance. This is because the variance of the value regressor in the GLM used to fit fMRI data depends on how much subjects learn about option values (no learning gives a flat regressor), unless regressors are z-scored (I could not find this info in the Materials and methods). This issue needs to be carefully addressed.*

Thank you for this comment, which is a valid concern. SPM by default mean centre the regressors. We have added a reference to that point in the Materials and methods:

“These regressors are mean-centered by default (20).”

To investigate the question in more detail we restricted our analysis to the high performers as defined by a median split (n=28, 13 old, 15 young). When we take this group, there is no longer an age group difference in total monetary gains (p=0.6). However, we found a correlation between Q in vmPFC and age (r(26)=-0.39; p=0.040) and a marginally significant group difference (t(26)=2.03; p=0.054). Therefore, it is unlikely that this difference comes only from the different performance in learning. This result has been added in the Results section:

“This difference in vmPFC value signal did not arise because of the difference in learning performance: when we restricted our analysis to high performers as defined by a median split (13 old, 15 young), a difference in performance was no longer significant (p=0.60), but the strength of expected-value signal in vmPFC was correlated with age (r(26)=-0.39, p=0.040) and we found a marginally significant difference between age groups (M_old_=4.21, SD=4.81; M_young_=8.29, SD=5.72; t(26)=2.03, p=0.054).”

*– The absence of (negative) correlation with expectation at outcome onset is interesting given the debate about prediction error encoding in the striatum. Yet I am unsure of how the authors interpret this. Is this an artifact from the design (cue and outcome onsets being too close in time), is it that true prediction errors are encoded in other brain regions, or is it that the brain does not encode prediction error at all? Perhaps the authors could clarify their position in this issue in the discussion.*

We have clarified our position in the Discussion. Although we would like to give a conclusive statement about whether and where the brain encodes RPEs, we do not think that our data can provide enough evidence one way or the other (Also see reviewer 2, point 6i).

“The lack of canonical RPE signal in NAcc could stem from the fact that we used a very stringent test for RPEs. Previous studies using the same stringent method report mixed results. Whereas some studies report significant positive effects of reward obtainment and negative effects of expected value (21,22), others do not find this canonical signal in NAcc (16,18,23,24). The conditions under which a canonical RPE can be detected may depend on task characteristics. For example, if the RPE signal is not behaviourally relevant for the task at hand it may not be encoded in the NAcc. In our case, however, RPEs are behaviourally relevant because the choice between bandits is based on fine-grained differences in their values. However, for other paradigms, the lack of behavioural relevance of RPEs could potentially explain a negative result (15,16,24). Another important aspect may be the temporal proximity of the choice cues and the outcome presentation in the task. This may hinder the dissection of opposing responses to these events with fMRI. We cannot rule out the possibility that our negative result stems from this feature of our task design and for this reason, we cannot provide conclusive evidence on the lack of canonical RPE signal in the NAcc. Our results point, however, to the need for stringent tests in future studies of the neural underpinnings of RPEs with fMRI.”

1. Lindenberger U, von Oertzen T, Ghisletta P, Hertzog C. Cross-sectional age variance extraction: what’s change got to do with it? Psychol Aging. 2011 Mar;26(1):34–47.

2. Dayan P, Sejnowski TJ. Exploration Bonuses and Dual Control. Mach Learn. 1996;25(1):5–22.

3. Badre D, Doll BB, Long NM, Frank MJ. Rostrolateral prefrontal cortex and individual differences in uncertainty-driven exploration. Neuron. 2012 Feb 9;73(3):595–607.

4. Wilson RC, Geana A, White JM, Ludvig EA, Cohen JD. Humans use directed and random exploration to solve the explore-exploit dilemma. J Exp Psychol Gen. 2014 Dec;143(6):2074–81.

5. Symmonds M, Wright ND, Bach DR, Dolan RJ. Deconstructing risk: separable encoding of variance and skewness in the brain. NeuroImage. 2011 Oct 15;58(4):1139–49.

6. Payzan-LeNestour E, Bossaerts P. Risk, unexpected uncertainty, and estimation uncertainty: Bayesian learning in unstable settings. PLoS Comput Biol. 2011 Jan 20;7(1):e1001048.

7. d’Acremont M, Fornari E, Bossaerts P. Activity in Inferior Parietal and Medial Prefrontal Cortex Signals the Accumulation of Evidence in a Probability Learning Task. PLoS Comput Biol [Internet]. 2013 [cited 2017 May 29];9(1). Available from: http://journals.plos.org/ploscompbiol/article?id=10.1371/journal.pcbi.1002895

8. Schönberg T, Daw ND, Joel D, O’Doherty JP. Reinforcement Learning Signals in the Human Striatum Distinguish Learners from Nonlearners during Reward-Based Decision Making. J Neurosci. 2007 Nov 21;27(47):12860–7.

9. Rutledge RB, Lazzaro SC, Lau B, Myers CE, Gluck MA, Glimcher PW. Dopaminergic Drugs Modulate Learning Rates and Perseveration in Parkinson’s Patients in a Dynamic Foraging Task. J Neurosci. 2009 Dec 2;29(48):15104–14.

10. Behrens TEJ, Woolrich MW, Walton ME, Rushworth MFS. Learning the value of information in an uncertain world. Nat Neurosci. 2007 Sep;10(9):1214–21.

11. Mathys C, Daunizeau J, Friston KJ, Stephan KE. A bayesian foundation for individual learning under uncertainty. Front Hum Neurosci. 2011;5:39.

12. Daw ND, Gershman SJ, Seymour B, Dayan P, Dolan RJ. Model-based influences on humans’ choices and striatal prediction errors. Neuron. 2011 Mar 24;69(6):1204–15.

13. McClure SM, Daw ND, Montague PR. A computational substrate for incentive salience. Trends Neurosci. 2003 Aug;26(8):423–8.

14. O’Doherty JP, Dayan P, Friston K, Critchley H, Dolan RJ. Temporal Difference Models and Reward-Related Learning in the Human Brain. Neuron. 2003 Apr 24;38(2):329–37.

15. Guitart-Masip M, Huys QJM, Fuentemilla L, Dayan P, Duzel E, Dolan RJ. Go and no-go learning in reward and punishment: Interactions between affect and effect. NeuroImage. 2012 Aug 1;62(1):154–66.

16. Stenner M-P, Rutledge RB, Zaehle T, Schmitt FC, Kopitzki K, Kowski AB, et al. No unified reward prediction error in local field potentials from the human nucleus accumbens: evidence from epilepsy patients. J Neurophysiol. 2015 Aug;114(2):781–92.

17. Rieckmann A, Karlsson S, Karlsson P, Brehmer Y, Fischer H, Farde L, et al. Dopamine D1 receptor associations within and between dopaminergic pathways in younger and elderly adults: links to cognitive performance. Cereb Cortex N Y N 1991. 2011 Sep;21(9):2023–32.

18. Chowdhury R, Guitart-Masip M, Lambert C, Dayan P, Huys Q, Düzel E, et al. Dopamine restores reward prediction errors in old age. Nat Neurosci. 2013 May;16(5):648–53.

19. Bari A, Robbins TW. Inhibition and impulsivity: Behavioral and neural basis of response control. Prog Neurobiol. 2013 Sep;108:44–79.

20. Mumford JA, Poline J-B, Poldrack RA. Orthogonalization of Regressors in fMRI Models. PLOS ONE. 2015 Apr 28;10(4):e0126255.

21. Behrens TEJ, Hunt LT, Woolrich MW, Rushworth MFS. Associative learning of social value. Nature. 2008 Nov 13;456(7219):245–9.

22. Niv Y, Edlund JA, Dayan P, O’Doherty JP. Neural prediction errors reveal a risk-sensitive reinforcement-learning process in the human brain. J Neurosci Off J Soc Neurosci. 2012 Jan 11;32(2):551–62.

23. Wimmer GE, Braun EK, Daw ND, Shohamy D. Episodic Memory Encoding Interferes with Reward Learning and Decreases Striatal Prediction Errors. J Neurosci. 2014 Nov 5;34(45):14901–12.

24. Li J, Daw ND. Signals in Human Striatum Are Appropriate for Policy Update Rather than Value Prediction. J Neurosci. 2011 Apr 6;31(14):5504–11.